# SARS-CoV-2 mRNA vaccines decouple anti-viral immunity from humoral autoimmunity

Jillian R. Jaycox[1,12], Carolina Lucas [1,12], Inci Yildirim [2,3,12], Yile Dai[1], Eric Y. Wang [1], Valter Monteiro [1], Sandra Lord[4], Jeffrey Carlin[5], Mariko Kita[5], Jane H. Buckner[6], Shuangge Ma[7], Melissa Campbell[8], Albert Ko [8,9], Saad Omer [3,8,9], Carrie L. Lucas [1], Cate Speake [4] ✉, Akiko Iwasaki [1,10] ✉ & Aaron M. Ring [1,11] ✉

mRNA-based vaccines dramatically reduce the occurrence and severity of COVID-19, but are associated with rare vaccine-related adverse effects. These toxicities, coupled with observations that SARS-CoV-2 infection is associated with autoantibody development, raise questions whether COVID-19 vaccines may also promote the development of autoantibodies, particularly in autoimmune patients. Here we used Rapid Extracellular Antigen Profiling to characterize self- and viral-directed humoral responses after SARS-CoV-2 mRNA vaccination in 145 healthy individuals, 38 patients with autoimmune diseases, and 8 patients with mRNA vaccine-associated myocarditis. We confirm that most individuals generated robust virus-specific antibody responses post vaccination, but that the quality of this response is impaired in autoimmune patients on certain modes of immunosuppression. Autoantibody dynamics are remarkably stable in all vaccinated patients compared to COVID-19 patients that exhibit an increased prevalence of new autoantibody reactivities. Patients with vaccine-associated myocarditis do not have increased autoantibody reactivities relative to controls. In summary, our findings indicate that mRNA vaccines decouple SARS-CoV-2 immunity from autoantibody responses observed during acute COVID-19.

mRNA-based vaccines against SARS-CoV-2 have demonstrated remarkable efficacy in preventing symptomatic infection and COVID-19 disease severity[1,2], even in the context of highly transmissible variants and sub-variants[3]. The brisk immune responses elicited by these vaccines are often accompanied by systemic inflammatory responses that include elevations of circulating plasma concentrations of cytokines such as IL-15, IFNγ, and CXCL10[4]. Though their overall safety compares favorably to other FDA-approved vaccines, adverse events ranging from common flu-like symptoms to rare cases of myocarditis have been observed[5]. Additionally, these vaccines have been associated with disease flares in patients with pre-existing autoimmune conditions[6].

[1]Department of Immunobiology, Yale School of Medicine, New Haven, CT, USA. [2]Department of Pediatrics, Section of Infectious Diseases and Global Health, Yale University School of Medicine, New Haven, CT, USA. [3]Yale Institute for Global Health, Yale University, New Haven, CT, USA. [4]Center for Interventional Immunology, Benaroya Research Institute at Virginia Mason, Seattle, WA, USA. [5]Virginia Mason Medical Center, Seattle, WA, USA. [6]Translational Research Program, Benaroya Research Institute at Virginia Mason, Seattle, WA, USA. [7]Department of Biostatistics, Yale School of Public Health, New Haven, CT, USA. [8]Department of Medicine, Section of Infectious Diseases, Yale University School of Medicine, New Haven, CT, USA. [9]Department of Epidemiology of Microbial Diseases, Yale School of Public Health, New Haven, CT, USA. [10]Howard Hughes Medical Institute, Chevy Chase, MD, USA. [11]Department of Pharmacology, Yale School of Medicine, New Haven, CT, USA. [12]These authors contributed equally: Jillian R. Jaycox, Carolina Lucas, Inci Yildirim. ✉e-mail: cspeake@benaroyaresearch.org; akiko.iwasaki@yale.edu; aaron.ring@yale.edu

Previously, we and others found that acute SARS-CoV-2 infection was associated with elevation of autoantibody reactivities, which also correlated with disease severity[7,8]. While some of these reactivities were likely preexisting, others were new or displayed an increasing trajectory that coincided with infection onset. The etiology of these antibodies has yet to be fully determined, and potential mechanisms intrinsic to the SARS-CoV2 spike protein—such as 'molecular mimicry'—raise the potential that vaccines targeting the same antigen could also drive humoral autoimmunity. Furthermore, whether autoantibody responses after vaccination differ in individuals previously infected with COVID-19 relative to naïve individuals has not yet been investigated.

In this work, we use REAP, an exoproteome-wide autoantibody screening platform, to show that autoantibodies are stable during vaccination relative to acute COVID-19, which is characterized by an elevated prevalence of new and increased autoantibody reactivities.

## Results and discussion

We serially monitored autoantibody and SARS-CoV-2-specific antibody responses of three separate cohorts pre- and post-vaccination (Fig. 1A). The first cohort[9] was composed of 33 healthcare workers (HCW) from Yale New Haven Hospital (YNHH), with approximately half the individuals seropositive for SARS-CoV-2 (Supplementary Table 1). The second cohort was composed of 38 individuals with pre-existing autoimmune diseases and 25 matched healthy controls recruited by the Benaroya Research Institute (BRI) (Supplementary Table 2). The autoimmune disease cohort was diverse, including 13 patients with Multiple Sclerosis (MS), 13 patients with rheumatoid arthritis (RA), 3 patients each with systemic lupus erythematosus (SLE), type 1 diabetes (T1D), and Crohn's disease (CD). The third cohort was comprised of 87 volunteers from the Dominican Republic who had previously received a two-dose course of the inactivated whole virion vaccine CoronaVac at least 4 weeks prior (Supplementary table 3). In this cohort, the mRNA vaccine was administered as a "booster." To assess longitudinal autoantibody dynamics in the absence of vaccination, we also included a group of 26 individuals monitored over time. This longitudinal control cohort was comprised of 14 healthy patients and 12 patients with type 1 diabetes (Supplementary Table 4).

To measure antibody responses against extracellular antigens and the SARS-CoV-2 S1 receptor binding domain (SARS-CoV-2-RBD), we used rapid extracellular antibody profiling (REAP), a yeast-display library platform that allows for simultaneous assessment of antibody reactivity against 6183 human extracellular proteins, peptide epitopes, and common coronavirus spike RBD proteins, including SARS-CoV-2 RBD[7,10]. Antibody reactivities detected by REAP are quantified by a REAP score (see Methods), which strongly correlates with antibody titers.

As anticipated, all individuals in the HCW cohort were reactive to the SARS-CoV-2-RBD by REAP after vaccination (Fig. 1B), with all seronegative individuals generating a new response. SARS-CoV-2-RBD REAP scores for seropositive individuals either remained stably high or increased after vaccination. Within the CoronaVac booster cohort, responses were more variable, but largely either increased or remained stable (Fig. 1C). Within the BRI cohort, all healthy individuals and most autoimmune patients responded to vaccination, showing reactivity to the SARS-CoV-2-RBD at the final time point (Fig. 1D). Patients on anti-CD20 B cell depletion therapy, the majority of whom are diagnosed with Multiple Sclerosis (MS), had significantly lower SARS-CoV-2-RBD antibody responses after vaccination ($p < 0.0001$) (Fig. 1E, S1A). Within this treatment group, MS patients taking Ocrelizumab were less likely to generate a humoral response than patients with other autoimmune diseases taking Rituximab (Fig. S1B). While most MS patients did not generate a SARS-CoV-2-RBD antibody response, two patients—one not on any therapy and one taking dimethyl fumarate (DMF)—generated normal SARS-CoV-2-RBD antibody responses by REAP (Fig. S1C).

To characterize the SARS-CoV-2 immune response amongst autoimmune and healthy individuals in the BRI cohort further, we performed ELISA for the S1 RBD and the full spike protein (S total) as well as neutralization assays against SARS-CoV-2 strain USA-WA1/2020. We found a strong correlation ($R = 0.9$, $p < 2.2e{-}16$) between the SARS-CoV-2-RBD REAP score and the S1 RBD ELISA titer (Fig. S1D), supporting the conclusions above generated from REAP. Interestingly, while most autoimmune patients produced a normal response to vaccination as measured by anti-S1 RBD titer or REAP score, these patients also exhibited decreased neutralization ability (PRNT50) (Fig. 1F). This trend was particularly pronounced for patients on anti-CD20 B cell depletion ($p < 0.0001$). Figure 1G shows S1 RBD ELISA titer and PRNT50 for all patients, with a linear regression for the relationship between these two parameters in healthy individuals only. A subset of patients falling to the lower right of the regression line, for example, several patients taking anti-TNFα and/or disease-modifying anti-rheumatic drug (DMARD) therapy, displayed high anti-S1 RBD titers with poor neutralization ability, indicating anti-RBD titer may not be entirely reflective of the quality of humoral immune protection in patients on immunosuppressive therapy. Finally, some patients who did not generate an S1 RBD-specific response still displayed reactivity to S total by ELISA, further supporting a constrained humoral response that may lead to targeting of non-RBD epitopes (Fig. 1H).

Next, we used REAP to investigate changes in autoreactivity to extracellular antigens during vaccination. While patients with autoimmune disease had a wider range in the number of preexisting autoantibody reactivities before vaccination, the mean number of reactivities did not significantly differ between controls or individuals with autoimmune disease, nor did they differ amongst the various autoimmune diagnoses (Fig. S2A, S2B). Similarly, in the Yale HCW cohort and the CoronaVac booster cohort, there was no difference in the number of preexisting reactivities between those who had previously been infected with SARS-CoV2 and those who were naïve (Fig. S2C, S2D). Overall, we found that the vast majority of autoantibody reactivities were remarkably stable over time in both the vaccination and longitudinal control cohorts. For example, Fig. 2A shows the autoantibody and SARS-CoV-2-RBD antibody trajectory of a RA patient (on anti-TNFα therapy). For this individual, autoantibodies detected by REAP were stable in the score during vaccination, except for anti-GPC6, which decreased over time. To display all individuals in each cohort overlaid, we normalized all REAP antigen scores to a starting score of 0 by subtracting their score at the first time point from their score at subsequent time points. Figure 2B shows the normalized REAP score trajectories for autoantibodies in every individual from the BRI cohort. Overall, autoantibody REAP score changes during vaccination were centered around zero (99.9% CI: Autoimmune: −0.10 to 0.29; Control: −0.043 to 0.56) and did not differ between patients with autoimmune disease and healthy controls.

The autoantibody dynamics of individuals in the absence of vaccination were also largely stable over time, with a similar degree of variation compared to the vaccination cohort (99.9% CI: Autoimmune: −0.62 to 0.28; Healthy: −0.37 to 0.38) (Fig. S3A). The average autoantibody change per individual (Fig. 2C) was also not different between healthy or autoimmune vaccinated patients, nor was there a difference between vaccinated and unvaccinated patients of both groups ($p = 0.075$). The similarity between autoimmune and healthy patients remained even after comparing the autoimmune patients on and off immunosuppression separately, suggesting that the lack of new autoantibodies in this group is not due to immunosuppression (Fig. S3B). Additionally, four RA patients had also received glucocorticoids during or surrounding their vaccination course. We did not note any differences in the average autoantibody change per individual between RA patients on vs. off glucocorticoids (Fig. S3C). Similar trends were observed in the CoronaVac booster cohort (99.9% CI: −0.71 to 0.16) (Fig. S3D) and the Yale HCW cohort (Fig. S3E); however,

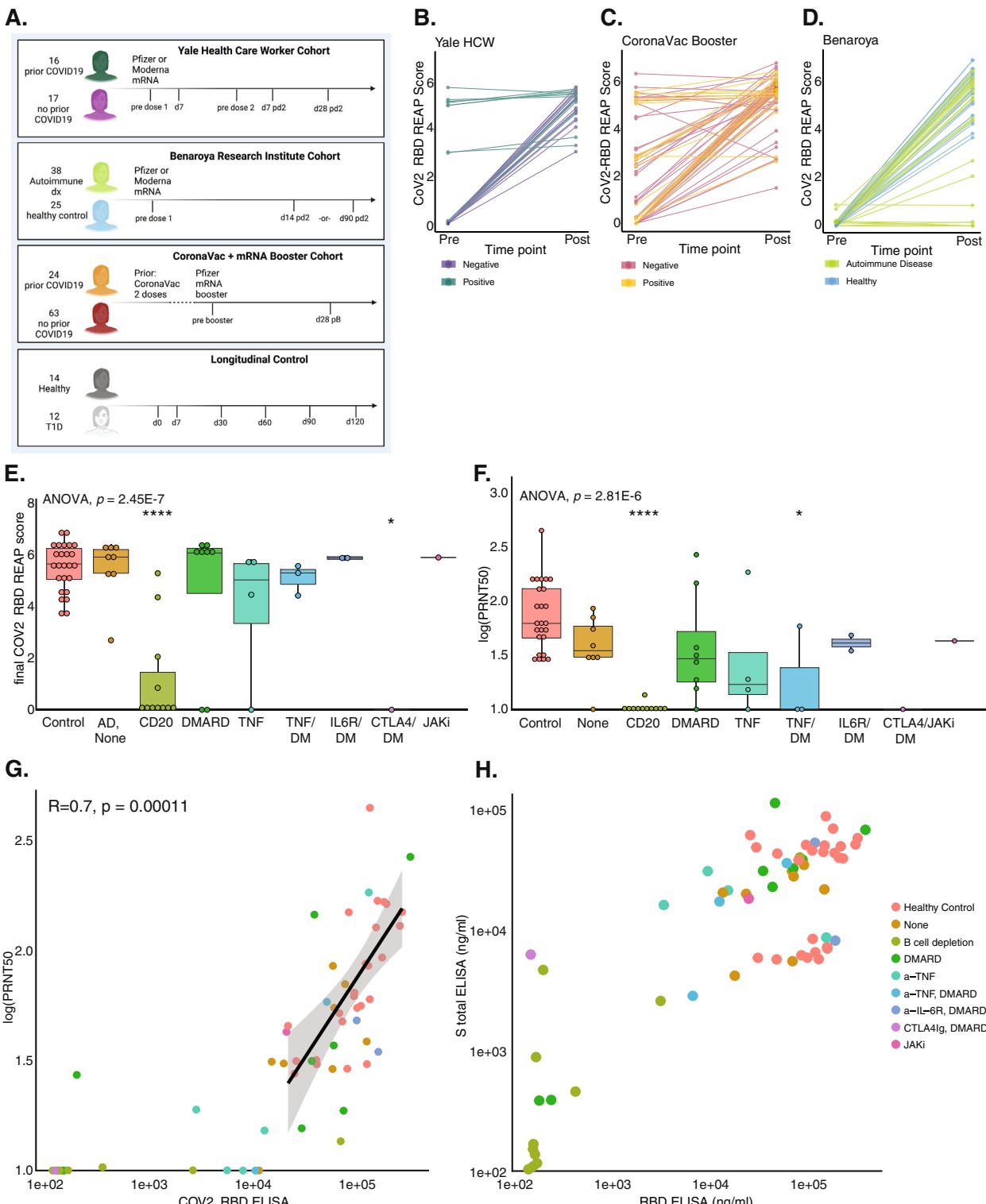

overall autoantibody changes were slightly negative for both HCW groups (99.9% CI: seronegative: −0.50 to −0.22; seropositive: −0.36 to −0.02). The average autoantibody change per individual did not differ between seronegative or seropositive healthcare workers or unvaccinated controls ($p = 0.69$) (Fig. S3F).

We next asked whether the autoantibody trajectories observed during SARS-CoV-2 vaccination differed from those observed during acute COVID-19. To answer this question, we analyzed REAP data from a cohort of 36 moderate and 23 severe COVID-19 patients who were hospitalized at YNHH between March and May of 2020[11]

(Supplementary Table 5). Visually, the autoantibody trajectories during vaccination differ strikingly from those observed in patients with moderate or severe acute COVID-19, who display numerous increased and new autoantibody reactivities over the course of infection (Fig. 2D). Nearly half of severe and a third of moderate COVID-19 patients had at least one increased or new autoantibody reactivity, with a substantial portion having multiple. (Fig. 2E, S4A, respectively). Conversely, this phenomenon was exceedingly rare during vaccination; out of 1034 total autoantibody reactivities detected amongst the vaccine cohorts, only 15 (1.45%) newly arose in the months after

**Fig. 1 | SARS-CoV-2-specific antibody response after mRNA vaccination of healthy individuals and autoimmune patients.** **A** Yale HCW: 2 doses mRNA vaccine; plasma collected at: pre dose 1–just prior to vaccination; d7–7 days post dose 1; pre dose 2–just prior to dose 2, approximately 21–28 days post dose 1; d7 pd2–day 7 post dose 2; d28 pd2–day 28 post dose 2. BRI: 2 doses mRNA vaccine; plasma collected at pre dose 1–prior to vaccination; d14 pd2–day 14 post dose 2; d90 pd2–day 90 post dose 2. CoronaVac Booster: 1 dose mRNA vaccine >4 weeks after 2 doses CoronaVac. Plasma collected at pre-Booster–prior to vaccination; d28–day 28 post booster. Longitudinal Control cohort: Unvaccinated participants; plasma collected at d0–day 0; d7–day 7; d30–day 30; d60–day 60; d90–day 90; d120–day 120 **B**–**D** CoV-2-RBD REAP score for each patient pre-vaccine to post-vaccine in the Yale HCW, CoronaVac Booster, and BRI cohorts, respectively. Each pair of dots represents a single individual. **E**, **F** CoV-2-RBD REAP score (**E**) and SARS-CoV-2 neutralization capacity (**F**) at the final time point. Significance was assessed by one-way ANOVA ($p = 2.45E - 7$ (**E**), $p = 2.81E - 6$ (**F**), with post hoc testing performed using Dunnett's test to compare the mean of every column with healthy control (CD20 vs. Control: $p = 5.1E - 8$ (**E**), CTLA4 vs. Control: $p = 0.0145$ (**E**); CD20 vs. Control: $p = 1.35E - 7$ (**F**), TNF/DM vs. Control: $p = 0.034$ (**F**). Boxplot box depicts the 25th to 75th percentile of the data, the middle line represents the median, and upper/lower whiskers represent max/min value within $1.5 \times 75th/25th$ interquartile range, respectively. $N = 25$ healthy individuals, 38 autoimmune individuals. **G** Neutralization capacity versus S1 RBD ELISA reactivity. Regression depicts the relationship between log(PRNT50) and S1 RBD ELISA for healthy participants, 95% C.I. shaded, regression line centered. Each dot represents a single individual. $N = 25$ healthy individuals, 38 autoimmune individuals. **H** S total ELISA reactivity (ng/ml) vs. S1 RBD ELISA reactivity (ng/ml), stratified by autoimmune disease status and medication category. ****$p < 0.0001$, ***$p < 0.001$ **$p < 0.01$ and *$p < 0.05$.

vaccination (Fig. S4B). For COVID-19 patients, we detected 24 new autoantibody reactivities out of 463 total autoantibodies (5.18%) (Fig. S4C). This metric is likely an underestimate of the increase in autoantibodies with COVID-19, due to the smaller library of antigens tested in the acute COVID-19 cohort (2777 antigens–Exo201 library) compared to the larger, updated library used for the vaccine cohort (6183 antigens) and the shorter time window in which these individuals were monitored. Similarly, the lack of a pre-infection baseline for these patients precludes the detection of new autoantibody reactivities that arose after infection, but before the first sample was collected for each patient. Overall, while autoantibody reactivities were stable in the vaccine cohort, we captured an increase in REAP score for TNFα in one patient with Rheumatoid Arthritis who was started on adalimumab therapy (an anti-TNFα monoclonal antibody) in the interval between the pre and post-vaccination samples (Fig. S4D). This finding demonstrates the sensitivity of REAP to detect new autoantibodies over time.

To account for temporal differences between the acute COVID-19 cohort, which averaged 10.3 (first time point) to 18.1 (final time point) days from symptom onset, and the vaccination cohorts, which are obligatorily at least 28 days long, but extended up to 4 months post dose 2 in some cases, we also performed a separate analysis on time points 28 days or less from dose 1 or symptom onset. In this temporally-matched analysis, the conclusions above are unchanged, with acute COVID-19 patients again having significantly more increased (Fig. S5A) and new (Fig. S5B) autoantibody reactivities and a significantly increased overall autoantibody delta (Fig. S5C).

To better understand the factors associated with the magnitude of increased autoantibody reactivities in COVID-19 patients compared to vaccinated patients, we performed multiple linear regression to model this phenomenon. In the model for the COVID-19 cohort (adjusted $R^2$: 0.1785, $p$ value: 0.015), increasing age, female sex, and clinical severity score of 6 were significantly associated with an elevated magnitude of increased autoantibody reactivities ($p = 0.0482$, $p = 0.0185$, $p = 0.0143$, respectively) (Fig. S6A–D). Furthermore, relative to a base model accounting for individual differences in age and sex only, the addition of clinical score significantly improved the goodness of fit of the overall model ($p = 0.01328$). Conversely, in the Benaroya vaccine cohort model (adjusted $R^2$: 0.0049, $p$ value: 0.39), neither sex, age, time span of monitoring, vaccine type, nor disease status (healthy vs. autoimmune) were significantly associated with the magnitude of increased autoantibody reactivities (Fig. S6E). Taken together, the comparison of autoantibody trajectories between COVID-19 and SARS-CoV-2 vaccination shows that COVID-19 patients display a characteristic pattern of new and increased autoantibodies that correlates with the highest clinical severity score, age, and female sex. Conversely, changes in autoantibodies during vaccination were similar to unvaccinated controls without any discernible pattern in new or increased reactivities, even in autoimmune-prone individuals.

Finally, we sought to profile extracellular autoantibodies in patients with mRNA vaccine-associated myocarditis, a rare but

potentially serious adverse event reported in association with SARS-CoV-2 mRNA vaccination[5]. To investigate this, we performed REAP on plasma samples from 8 patients presenting to YNHH between May to October 2021 with MRI-verified myocarditis beginning 1–3 days (mean: 2.75) after the second mRNA vaccine dose (Supplementary table 6). The myocarditis cohort exclusively received the Pfizer vaccine, as this was the only approved mRNA vaccine at the time of our study. Figure 2F shows a heatmap of REAP autoantibody reactivity scores for myocarditis patients and age and sex-matched controls. Myocarditis patients did not display elevated numbers of autoantibodies compared to controls (Fig. 2G), nor did they display autoantibodies against cardiac or endothelial enriched antigens or antigens previously implicated in cardiac inflammatory disorders, such as anti-B adrenoreceptor antibodies[12]. One study recently reported a high frequency of functional anti-IL1RA autoantibodies in patients with vaccine-associated myocarditis[13]. Interestingly, we did not detect IL-1RA autoantibodies in our cohort by REAP or IL-1RA ELISA (Fig. 2H, S7A). This discrepancy may be due to differences in cohort demographics or a high frequency of anti-drug antibodies against IL-1RA biologic agents in the other report, which did not note whether patients had received this therapy.

Our findings here fit with several recent reports highlighting innate immune activation and increased frequencies of activated cytotoxic CD8 T cells and NK cells in vaccine-associated myocarditis, as opposed to B cell activation or plasma cell infiltration[14–16]. However, due to the limitations of yeast display technology, the REAP library does not include intracellular proteins. Therefore, our findings do not eliminate the possibility that there may be intracellularly targeted autoantibodies, for example, anti-myosin, which has also been previously reported in other types of myocarditis. Overall, however, our results show that changes in autoantibodies specific for extracellular antigens are unlikely to underlie mRNA vaccine-associated myocarditis.

While we did detect a small number of new or increased autoantibody reactivities that occurred after vaccination, the presence of this phenomenon in the unvaccinated control group suggests that this may be due to physiological variation in autoantibody concentrations rather than an effect of vaccination. Furthermore, several factors argue against causally linked or stereotypical autoantibody responses in vaccinated patients. These include the lack of any difference between autoantibody changes in autoimmune versus healthy patients; the fact that net autoantibody changes were centered around zero (i.e., approximately the same number of autoantibody responses increased after vaccination as those that decreased); and the lack of shared autoantibodies amongst patients with or without mRNA vaccine-associated myocarditis.

Additionally, the lack of autoantibody changes observed after vaccination contrasted markedly with acute COVID-19, which is associated with increased and new autoantibody reactivities arising during the course of the disease. This finding fits with an emerging picture of pathological humoral immune dysfunction in COVID-19, leading to

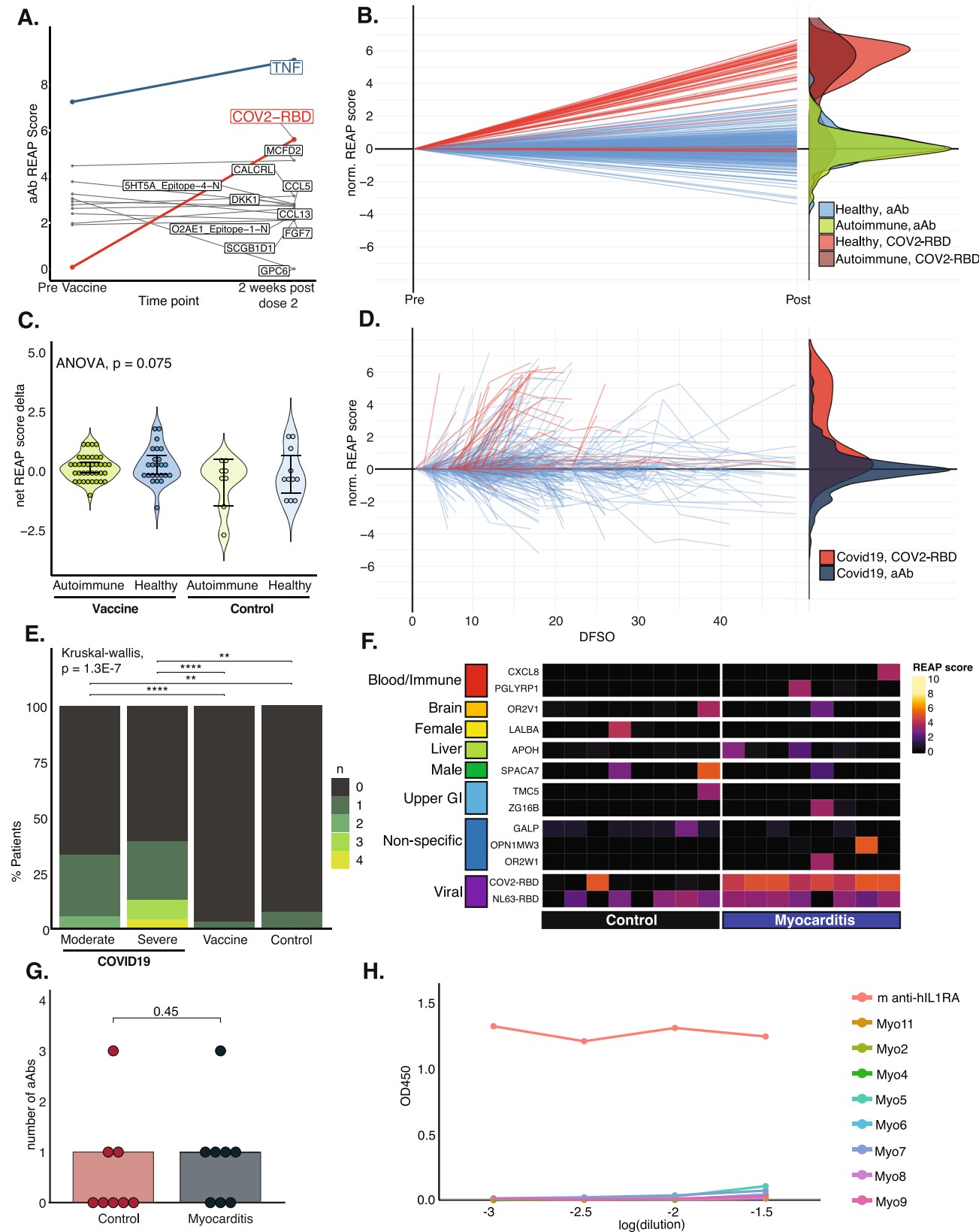

exuberant B cell responses reminiscent of systemic lupus erythematosus (SLE)[17] and production of autoantibodies that cause thrombotic syndromes[18], dampen type 1 IFN programs[19,20], and neutralize cytokine and chemokine signaling[7]. While the mechanisms behind the generation of these autoantibodies are unclear, it is plausible that pathological inflammation plays a role, for example leading to tissue damage and exposure of sequestered antigens, bystander activation of autoreactive B cell clones by a hyperinflammatory cytokine milieu, and/or exaggerated proliferation of polyspecific B cell clones. The difference in autoantibody dynamics between acute COVID-19 and vaccination and the correlation of the highest COVID-19 clinical severity score with the magnitude of increased reactivities further support the idea that

**Fig. 2 | Autoantibody dynamics during mRNA vaccination and acute COVID-19.**
**A** Antibodies of an RA patient during mRNA vaccination (CoV-2-RBD−red, auto-antibody−gray, therapeutic a-TNF Ab−blue). **B** *Line plot*: autoantibody (blue) and CoV-2 RBD (red) REAP score trajectories of the BRI cohort during vaccination (Exo201 antigens only). Each line represents one reactivity normalized to a baseline score of 0. *Density plot*: REAP score deltas for reactivities in the BRI cohort. Probable drug antibodies (a-TNF, a-IL6R) were excluded. **C** Average REAP score change for autoantibody reactivities per individual from the first to the final time point in the BRI and longitudinal control cohorts (all antigens). $p = 0.075$, by one-way ANOVA. Error bars show 99% CI. Probable drug antibodies (a-TNF, a-IL6R) were excluded. Each dot represents one individual. Individuals with zero reactivities were excluded. $N$ = vaccine/autoimmune: 37; vaccine/healthy: 24; control/auto-immune: 8; control/healthy:11. **D** *Line plot*: autoantibody (blue) and CoV-2 RBD (red) REAP score trajectories of patients with acute COVID-19 (Exo201 antigens only). Each line represents one reactivity normalized to a starting score of 0. DFSO = days from symptom onset. *Density plot*: REAP score deltas for reactivities in

the COVID-19 cohort. Probable drug antibodies (a-TNF, a-IL6R) were excluded. **E** Proportion of patients with $n$ increased reactivities (Exo201 antigens only). $p = 1.3E − 7$, by Kruskal−Wallis, Dunn's post hoc test (correction by Holm's method): severe vs. vaccine, $p = 6.1E − 7$; severe vs. control, $p = 1.1E − 3$; moderate vs. vaccine, $p = 6.1E − 7$; moderate vs. control, $p = 4.0E − 3$. Increased reactivity = increase in REAP score by >3 at any time point. Probable drug antibodies (a-TNF, a-IL6R) were excluded. N: severe COVID19: 23; moderate COVID19: 36; Vaccine: 183; Control: 26. **F** Reactivities in vaccine-associated myocarditis patients or controls. Antigens grouped by Human Protein Atlas expression data. **G** Autoantibody reactivities per individual in mRNA vaccine-associated myocarditis cohort vs. control. $p = 0.45$, unpaired two-sided $t$-test. Boxplot box depicts the 25th to 75th percentile of the data, the middle line represents the median, and upper/lower whiskers represent the max/min value within the 1.5× 75th/25th interquartile range, respectively. N = Control: 8; Myocarditis: 8. **H** IL-1RA autoantibody ELISA of myocarditis patient serum. $****p < 0.0001, ***p < 0.001 **p < 0.01$ and $*p < 0.05$.

autoantibody induction is related to the immunopathology of COVID-19, rather than attributes of the SARS-CoV-2 spike antigen leading to phenomena such as 'molecular mimicry'.

Citing increases in breakthrough infections, the CDC continues to recommend a third and fourth[21] mRNA vaccine dose for immuno-compromised individuals, including autoimmune disease patients on immunosuppressive therapies. This subject is the current focus of several ongoing clinical trials. While our study is limited in sample size for certain diagnosis and medication groups, it highlights a limitation of relying on anti-SARS-CoV-2 titers as a correlate of protective immunity in autoimmune/immunosuppressed patients. Notably, we found that although most autoimmune patients generate RBD-specific antibodies after vaccination, their antibodies were frequently non-neutralizing, unlike healthy individuals, where the correlation between antibody titers and neutralizing ability was strong. This suggests that the anti-SARS-CoV-2 antibody response generated in these individuals may be constrained and less effective, for example, due to targeting of non-critical epitopes or lack of protective anti-N-terminal-domain (NTD) antibodies[22]. This was particularly evident for individuals on B cell depletion therapies, who were the least likely to generate a neu-tralizing antibody response, in accordance with another recent report[23]. Interestingly, some of these patients nevertheless generated non-neutralizing antibodies against the full S protein, which likely reflects an inefficient humoral response constrained by B cell deple-tion. It is important to note, however, that our study did not measure T cell responses to SARS-CoV-2 vaccination, which have been shown to be an important contributor to SARS-CoV-2-specific immunity[24,25]. This is particularly relevant for individuals on immunosuppression, where humoral and cellular immune protection after vaccination may not be expected to correlate, as has been observed with MS patients on Ocrelizumab[23]. Therefore, we cannot definitively infer the full degree of immune protection for individuals in our study.

In conclusion, we found that mRNA vaccination was not asso-ciated with the development of new autoantibody responses to the extracellular proteome in stark contrast to SARS-CoV-2 infection. Furthermore, despite differences in the quality of the SARS-CoV-2 response elicited in autoimmune and healthy patients after mRNA vaccination, there was no difference in the dynamics of autoantibodies against extracellular self-antigens, even in patients with preexisting autoimmune disease or vaccine-related myocarditis. This work thus bolsters the emerging safety profile of mRNA vaccines and highlights their ability to decouple SARS-CoV-2 immunity from the potential long-term autoimmune sequelae of COVID-19.

## Methods
### Study approvals
For the BRI cohort, subjects with and without autoimmunity were enrolled under protocol number IRB08108, approved by the BRI

Institutional Review Board. All subjects provided written informed consent to participate in this study per study protocol. For the Yale HCW cohort, subjects were enrolled under protocol number 2000028924, approved by the Yale Human Research Protection Pro-gram Institutional Review Board. Informed consent was obtained from all enrolled participants. The CoronaVac Booster study was approved by the National Bioethics Committee of the Dominican Republic (CONABIOS). The participants received two doses of the inactivated whole-virion vaccine CoronaVac followed by a BNT162b2 booster dose at least four weeks after the second dose of CoronaVac. All DR parti-cipants consented to enroll in this observational study. For the myo-carditis cohort, subjects were enrolled under protocol numbers 2000028924 and 1605017838 approved by the Yale Human Research Protection Program Institutional Review Board. Informed consent was obtained from all enrolled participants. All human subjects research described here was performed in accordance with the Declaration of Helsinki.

Sex was considered in the study design, with the goal of recruiting approximately an equal proportion of male and female participants, wherever possible. Sex was determined based on participant self-reporting.

### Classification of acute COVID-19 severity
Patients in the acute COVID-19 cohort were stratified by disease severity, based on oxygen levels and ICU requirements, as described previously[11]. Moderate disease status (Clinical Score 1, 2, or 3) was defined as: (1) SARS-CoV-2 infection requiring hospitalization without supplemental oxygen, (2) infection requiring non-invasive supple-mental oxygen (<3 L/min, sufficient to maintain >92% SpO2), or (3) infection requiring non-invasive supplemental oxygen (>3 L/min, suf-ficient to maintain >92% SpO2, or, required >2 L supplemental oxygen to maintain SpO2 > 92% and had a high-sensitivity C-reactive protein (CRP) > 70 and received Tocilizumab). Severe disease status (clinical score 4 or 5) was defined as meeting criteria for score 3 while also requiring admission to the YNHH intensive care unit (ICU) and >6 L supplemental oxygen to maintain SpO2 > 92% (4), or infection requir-ing invasive mechanical ventilation and/or extracorporeal membrane oxygenation (ECMO) in addition to glucocorticoid/vasopressor administration (5). Clinical score 6 was assigned for deceased patients; however, in this study, it is included in the severe disease group.

### Yeast library generation
**Library design.** The initial yeast library (Exo201) was generated as previously described[7,10]. In Exo201, only the largest extracellular domain of multi-pass membrane proteins was included in the library. In this study, we further expanded the library by supplementing with all extracellular domains of multi-pass membrane proteins greater than 15 amino acids. Additionally, we identified and added larger

antigens that failed in Exo201, for example, due to PCR failure. An inventory of newly added antigens is compiled in Supplementary dataset 1. DNA for new antigens was synthesized as either a Gene Fragment (for antigens over 300 nucleotides) or as an Oligo pool by TWIST Bioscience, containing a 5′ sequence (CTGTTATTGC TAGCGTTTTAGCA) and 3′ sequence (GCGGCCGCTTCTGGTGGC) for PCR amplification. The oligo pool was PCR amplified and transformed into yeast with barcode fragments, followed by barcode-antigen pairing identification as previously described[7,10]. This new yeast library was then pooled with the initial library (Exo201) in a ratio of 1:1 to generate the new version of the library (*Exo204*).

## Rapid extracellular antigen profiling

**Antibody purification and yeast adsorption.** Antibody purification was performed as previously described[7,10]. Briefly, triton x-100 and RNase were added to patient plasma at a final concentration of 0.5% and 0.5 mg/ml, respectively, and incubated for 30 min to inactivate enveloped RNA viruses. 20 μL protein G magnetic resin (lytic solutions) was washed and resuspended in PBS and added to 50 μL of inactivated plasma. Serum-resin mixture was incubated for three hours at 4 °C with shaking. The resin was washed with PBS and resuspended in 90 μL 100 mM glycine pH 2.7 for 5 min. The supernatant was extracted and added to 10 μL sterile 1 M Tris pH 8.0. Yeast adsorption was performed as previously described[7,10]. Briefly, empty vector (pDD003) yeast was induced by culture in 1:10 SDO-Ura:SGO-Ura for 18 h. $10^8$ induced yeast were washed with PBE (PBS with 0.5% BSA and 0.5 mM EDTA), resuspended with 100 μL purified IgG, and incubated for three hours at 4 °C with shaking. Yeast-depleted IgG was eluted from the Yeast-IgG mixture through 0.45 um filter plates by centrifugation at 3000*g* for 3 min.

**Patient-derived antibody yeast library selections.** Yeast library selections were performed as previously described[7,10]. Briefly, the Exo201 (acute COVID-19 cohort) or Exo204 (BRI, Yale HCW, CoronaVac, longitudinal control, and myocarditis cohorts) yeast library was induced at an OD of 1 cultured in 1:10 SDO-Ura:SGO-Ura at 30 C. Prior to selection, $5^8$ induced yeast were set aside to allow for comparison of the pre-selection to post-selection libraries. $10^8$ induced yeast were washed with PBE and added to wells of a sterile 96-well plate. Ten micrograms of yeast adsorbed IgG was added to the yeast library in duplicate in 100 uL PBE and incubated for 1 h at 4 C. Yeast were washed with PBE and incubated with 1:100 biotin anti-human IgG Fc antibody (clone HP6017, BioLegend #409308, or clone QA19A42, Biolegend #366918) for 30 min. Yeast was washed with PBE and incubated with a 1:20 dilution of Streptavidin MicroBeads (#130-048-101, Miltenyi Biotec) for 30 min. Yeast was resuspended in PBE and IgG-bound yeast was isolated by positive magnetic selection using the MultiMACS M96 Separator (Miltenyi Biotec) according to manufacturer instructions and as previously described[7,10]. Selected yeast was resuspended in 1 mL SDO -Ura and incubated at 30 °C for 24 h.

**Next-generation sequencing (NGS) library preparation and sequencing.** NGS library preparation was performed as previously described[7,10]. Briefly, DNA was extracted from yeast libraries using Zymoprep-96 Yeast Plasmid Miniprep kits or Zymoprep Yeast Plasmid Miniprep II kits (Zymo Research) according to standard manufacturer protocols. A first round of PCR was used to amplify a DNA sequence containing the protein display barcode on the yeast plasmid, as previously described[7,10]. A second round of PCR was performed on 1 μL step 1 PCR product using Nextera i5 and i7 dual-index library primers (Illumina), as previously described[7,10]. PCR products were pooled, and run on a 1% agarose gel, and DNA corresponding to the band at 257 base pairs was cut. DNA (NGS library) was extracted using a QIAquick Gel Extraction Kit (Qiagen) according to standard manufacturer protocols. NGS library was sequenced using an Illumina NextSeq550 and a NextSeq high output sequencing kit with 75 base pair single-end

sequencing according to standard manufacturer protocols. A minimum of 200,000 reads on average per sample was collected and the pre-selection library was sampled at least ten times greater depth than other samples. Samples with fewer than 50,000 reads were discarded as failed sequencing.

**Data analysis.** REAP scores were calculated as previously described[7,10]. Briefly, barcode counts were extracted from raw NGS data using custom codes and counts from technical replicates were summed. Next, aggregate and clonal enrichment were calculated using edgeR[26] and custom codes. *Aggregate enrichment* is the log2 fold change of all barcodes associated with a particular protein summed in the post-library relative to the pre-library, with zeroes in place of negative fold changes. Log2 fold change values for *clonal enrichment* were calculated in an identical manner, but barcode counts across all unique barcodes associated with a given protein were not summed. Clonal enrichment for a given reactivity was defined as the fraction of clones out of total clones that were enriched (log2 fold change ≥ 2). Aggregate (Ea) and clonal enrichment (Ec) for a given protein, a scaling factor (βu) based on the number of unique yeast clones (yeast that have a unique DNA barcode) displaying a given protein, and a scaling factor (βf) based on the overall frequency of yeast in the library displaying a given protein were used as inputs to calculate the REAP score, which is defined as follows.

$$REAP\ score = Ea \times (Ec)^2 \times \beta u \times \beta f$$

βu and βf are logarithmic scaling factors that progressively penalize the REAP score of proteins with low numbers of unique barcodes or low frequencies in the library, and are described in detail in previous publications[7,10].

Antigens with an average REAP score greater than 0.5 across all samples were defined as non-specific binders and excluded from further analysis. Autoantibody reactivities were defined as antigens with REAP score >2 and >1.96 row- (antigen-) wise *Z* score threshold (unless otherwise noted).

## Cell lines and virus

TMPRSS2-VeroE6 kidney epithelial cells were cultured in Dulbecco's Modified Eagle Medium (DMEM) supplemented with 1% sodium pyruvate (NEAA) and 10% fetal bovine serum (FBS) at 37 °C and 5% $CO_2$. The cell line has tested negative for contamination with mycoplasma. SARS-CoV-2 lineage A(USA-WA1/2020) was obtained from BEI Resources (#NR-52281) and was amplified in TMPRSS2-VeroE6. Cells were infected at an MOI 0.01 for 3 days to generate a working stock and after incubation, the supernatant was clarified by centrifugation (450*g* × 5 min), and filtered through a 0.45-μm filter. The pelleted virus was then resuspended in PBS and aliquoted for storage at −80 °C. Viral titers were measured by standard plaque assay using TMPRSS2-VeroE6. Briefly, 300 μl of serial fold virus dilutions were used to infect Vero E6 cells in MEM-supplemented NaHCO3, 4% FBS 0.6% Avicel RC-581. Plaques were resolved at 48 h post-infection by fixing in 10% formaldehyde for 1 h followed by 0.5% crystal violet in 20% ethanol staining. Plates were rinsed in water to allow for plaque enumeration. All experiments were performed in a biosafety level 3 laboratory with approval from the Yale Environmental Health and Safety office.

## Neutralization assay

Neutralization assay was conducted as described previously[9]. Briefly, sera from vaccinated individuals were heat treated for 30 min at 56 °C. Sixfold serially diluted plasma, from 1:10 to 1:2430 were incubated with SARS-CoV-2 lineage A(USA-WA1/2020) for 1 h at 37 °C. The mixture was subsequently incubated with TMPRSS2-VeroE6 in a 12-well plate for 1 h, for adsorption. Then, cells were overlaid with MEM-supplemented NaHCO3, 4% FBS 0.6% Avicel mixture. Plaques were resolved at 40 h

post-infection by fixing in 10% formaldehyde for 1 h followed by staining in 0.5% crystal violet. All experiments were performed in parallel with baseline control sera, in an established viral concentration to generate 60–120 plaques/well.

## SARS-CoV-2 specific-antibody measurements

ELISAs were performed as previously described[9]. Briefly, Triton X-100 and RNase A were added to serum samples at final concentrations of 0.5% and 0.5 mg/ml respectively, and incubated at room temperature (RT) for 30 min before use, to reduce risk from any potential virus in serum. 96-well MaxiSorp plates (Thermo Scientific #442404) were coated with 50 μl/well of recombinant SARS Cov-2 S Total (ACROBiosystems #SPN-C52H9-100 μg) or RBD (ACROBiosystems #SPD-C52H3-100 μg) protein at a concentration of 2 μg/ml in PBS and were incubated overnight at 4 °C. The coating buffer was removed, and plates were incubated for 1 h at RT with 200 μl of blocking solution (PBS with 0.1% Tween-20, 3% milk powder). Plasma was diluted serially at 1:100, 1:200, 1:400, and 1:800 in dilution solution (PBS with 0.1% Tween-20, 1% milk powder) and 100 μl of diluted serum was added for two hours at RT. Human Anti-Spike (SARS-CoV-2 Human Anti-Spike (AM006415) (Active Motif #91351) was serially diluted to generate a standard curve. Plates were washed three times with PBS-T (PBS with 0.1% Tween-20) and 50 μl of HRP anti-Human IgG Antibody (GenScript #A00166, 1:5000) diluted in dilution solution added to each well. After 1 h of incubation at RT, plates were washed six times with PBS-T. Plates were developed with 100 μl of TMB Substrate Reagent Set (BD Biosciences #555214) and the reaction was stopped after 5 min by the addition of 2 N sulfuric acid. Plates were then read at a wavelength of 450 and 570 nm.

## IL-1RA ELISA

ELISAs were performed as previously described[7]. Briefly, 96-well MaxiSorp plates (Thermo Scientific #442404) were coated with 200 ng of recombinant IL-1RA protein (Biolegend #553906) in PBS and incubated overnight at 4 C. Plates were dumped out and incubated with 2% Human Serum Albumin (HSA) (Celprogen #HSA2001-25-2) in PBS for 2 h at RT. Plates were washed 3× with 200 μl wash buffer (PBS 0.05% Tween). Samples were diluted in 2% HSA and added to the plate to incubate for 2 h at RT. Mouse anti-human IL-1RA (Prospec #ant-238) was used as a positive control. Plates were washed 6× with wash buffer. Goat anti-human IgG Fc (Sigma Aldrich, #AP113P) diluted 1:10000 in 2% HSA was added to the plates and incubated for 1 h at RT. For the positive control, 1:5000 Goat anti-mouse IgG Fc (Thermo Fisher Scientific, #A16088) in 2% HSA was used. Plates were washed 6x. Plates were developed with 100 μl of TMB Substrate Reagent Set (BD Biosciences #555214) and the reaction was stopped after 5 min by the addition of 2 N sulfuric acid. Plates were then read at a wavelength of 450 nm.

## Statistics and reproducibility

Statistical details of experiments can be found in the figure legends. All REAP screens, ELISA, and neutralization assays were performed with two technical replicates. Data analysis was performed using R, Python, Excel, and GraphPad Prism. No statistical method was used to pre-determine the sample size. Patients were excluded from the acute COVID-19 cohort if they did not have moderate or severe disease or if they only had a 1-time point because this would prohibit the longitudinal analysis required by our study. Patients were excluded from the vaccination cohort if an unclear sample label or obvious cross-well contamination was detected. Additionally, one patient was excluded due to failed REAP process and lack of autoantibody data. Two patients were excluded from the Myocarditis cohort: 1 patient received IVIG before their blood sample, which complicates the results of REAP, making them uninterpretable; 1 patient had onset of myocarditis 21 days after the vaccination, which is an atypical time course and thus

the cause of the myocarditis (viral vs. vaccine) could not be determined. The experiments were not randomized.

For the acute COVID-19 cohort and the Benaroya cohort, REAP and ELISA/neutralization studies were performed by the investigator before receiving associated clinical annotations. For the Yale HCW cohort, CoronaVac cohort, and the myocarditis cohort, the investigator received clinical annotations at the time of REAP/ELISA/neutralization studies. However, samples were analyzed in the same manner in a randomized plate layout.

### Reporting summary

Further information on research design is available in the Nature Portfolio Reporting Summary linked to this article.

## Data availability

All data reported in this paper will be shared by the corresponding author upon request. Any additional information required to reanalyze the data reported in this paper is available from the corresponding author upon request. Source data are provided in this paper.

## Code availability

The code used to generate the results and figures present in the manuscript will be shared by the corresponding author upon request.

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

## Acknowledgements

We thank Suzanne Fischer for logistical assistance and Jonathan Klein for helpful discussion. Rachel Hartley provided crucial assistance in sample and clinical data tracking for the BRI cohort. We thank Andrew Pickles, Heather White, Kassidy Benoscek, and Kimberly Varner for their work in recruitment, enrollment, and the conduct of BRI cohort study visits. Carla Greenbaum, MD, provided advice and consulting for the study, and Uma Malhotra, MD, provided oversight of the BRI IRB protocol. We also acknowledge the clinicians and staff of the Virginia Mason Franciscan Health Rheumatology Department for assistance in recruiting study participants for the BRI cohort. Figure 1A was created with BioRender.com. This work was supported by the Mathers Family Foundation (to A.M.R. and A.I.), the Ludwig Family Foundation (to A.M.R. and A.I.), and a supplement to the Yale Cancer Center Support Grant 3P30CA016359-40S4 (to A.M.R.). IMPACT received support from the Yale COVID-19 Research Resource Fund. The Benaroya Family Foundation, the Leonard and Norma Klorfine Foundation, Glenn and Mary Lynn Mounger, and Monolithic Power Systems provided financial support to BRI for this work. A.M.R. is additionally supported by an NIH Director's Early Independence Award (DP5OD023088) and the Robert T. McCluskey Foundation. J.R.J is supported by the Yale Medical Scientist Training Program T32GM007205. C.L. is a Pew Latin American Fellow. A.I. is an investigator at the Howard Hughes Medical Institute.

## Author contributions

J.R.J., C.L., Y.D., and V.M. performed experiments. I.Y. provided Yale HCW and myocarditis patient samples and clinical annotations. J.B., C.S., M.K., S.L., and J.C. provided BRI cohort samples and clinical annotations. J.R.J., E.Y.W., and C.L. analyzed data. S.M. provided statistics and data analysis supervision. A.K., M.C., S.O., C.S., A.I., and A.M.R. provided project supervision. J.R.J. and A.M.R. wrote the paper.

## Competing interests

E.Y.W., Y.D., and A.M.R. are inventors of a patent describing the REAP technology and A.M.R. is the founder of Seranova Bio. The remaining authors declare no competing interests.
