## [Peer review file · Nature Communications]

REVIEWER COMMENTS

Reviewer #1 (Remarks to the Author):

The article by Jaycox and al., uses REAP (Rapid Extracellular Antigen Profiling) to study antibody responses after vaccination with SARS-CoV2 mRNA vaccines.

They study relatively large cohorts of 145 healthy vaccinated individuals, in addition to 38 patients with autoimmune diseases, and 8 patients with mRNA vaccine-associated myocarditis.

The paper is interesting and relatively novel in the question posed, and the study of post-vaccine auto-Abs, as well as auto-Abs in the context of vaccine induced myocarditis.

Comments:

- It is at times a bit difficult to follow the different cohorts in the manuscript (Fig 1A is helpful though) and might benefit from simplifying the first result paragraphs where the cohorts are described sequentially.

- Are there any differences between Moderna and Pfizer mRNA vaccines?

- Details of the auto-immune's disease could be detailed in the manuscript (at least to mention the number of patients per disease).

- Is Crohns considered an auto-immune disease or an inflammatory disease? In addition, the detection of auto-Abs to TNFa is interesting but not relevant for the present study. Removing these patients could be considered.

- In figure 1B, C it would be helpful to split the figure in two, to really identify the vaccinated and unvaccinated individuals

- In the second paragraph, the authors should differentiate the auto-Abs that are secondary to the infection by SARS-CoV-2 to those that are pre-existing the infection and can cause severe disease (i.e. Abs to type I IFNs).

- It would be interesting to stratify per auto-immune disease, to see if there are common auto-Abs that persist. Also, does REAP detect the auto-Abs known to be associated to these AI diseases (especially T1D, RA...)

- In MS patients treated with Rituximab, do the authors have the information on the date of last injection or number of B cells? That would be of particular interest for the patient with a normal response.

- How can the authors explain a good response by ELISA or REAP titers, but a decreased neutralization capacity, in figure 2E?

- How would the authors explain fig 1G, showing a cross-reactivity to non-RBD epitopes in individuals that have only been vaccinated by mRNA vaccines against S protein? Sequence homology? Has this been described elsewhere?

- For the auto-Abs arising after COVID-19 infection, have the authors assessed their persistence over time?

- Do the authors find auto-Abs to type I IFNs in the patients infected with severe COVID-19. If yes, do they also detect them before infection?

- For patients with myocarditis, could some pathogenic auto-Abs be undetected by REAP, either because they are not included in the panel or because of a conformational issue? It might be important to discuss this.

- All the patients with myocarditis had been vaccinated with Pfizer. Do the authors have an explanation, based on their cohort recruitment? It should be mentioned in the text.

- The title is a bit ambiguous, as the reader cannot understand at a glance if there are more auto-Abs in vaccinated individuals or the opposite.

Reviewer #2 (Remarks to the Author):

In this manuscript, Jaybox et al. used REAP score, an assay that detects several thousands of extracellular antigens, to test the humoral responses in individuals vaccinated with mRNA vaccine, with or without autoimmune diseases, or myocarditis triggered by vaccination. They hypothesis tested in this paper is whether COVID mRNA vaccine triggers the production of autoantibodies in the recipients. Overall, I found the hypothesis is not clearly defined, the design of the study lacks rationale, the method and cohorts chosen are not appropriate to test the hypothesis.

1. The so called "autoimmune patients" is a massively heterogeneous cohort, including MS, SLE, T1D, etc. Lumping them together as one cohort ignores the fact that they have different autoantibodies, and different disease mechanisms.
2. Among the autoimmune patients tested in this paper, did any of them had autoimmunity flare or change of clinical diseases post vaccination? If not, it means that the autoimmune diseases in these patients are not associated with vaccination, therefore they are not the appropriate patients to study. Just because they have "autoimmune diseases", which again is not one well-defined disease, doesn't make them the correct patients to study for autoantibodies.
3. What does day 0 mean in the longitudinal cohort? The onset of T1D?
4. The autoimmune diseases the authors studied in this paper, are not characterized by specific functional autoantibodies like anti-ACHR in myasthenia gravis. The author did not show that the REAP assay can identify the characteristic autoantibodies such as anti-beta cell in T1D. What is the rationale for using the REAP assay, which contains several thousand antigens, to test the antibody response of mRNA vaccine which contains RNA that codes for a small peptide? I understand the rationale for using REAP assay in COVID patients, who encountered a new pathogen with a variety of new antigens, and actually leads to tissue damage that releases many self-antigens and PAMPs. But for mRNA vaccine, it is not surprising that a small peptide did not change the diversity of several thousands of autoantibodies in any of the cohorts the authors tested. Is there any evidence in the literature that a small peptide can change the diversity of autoantibody/antibody pool in vivo?
5. How is it possible that some patients generated non-RBD antibodies through vaccination (Fig 1G)? I think the better explanation for this data are: (1) these patients had COVID before and vaccination failed, therefore they have antibodies against S but not RBD; or (2) which is more likely, that the RBD-ELISA does not detect all anti-RBD antibodies, which is very common in ELISA-based assay especially with small peptides, as the coating of the peptide to the ELISA plates can selectively mask some epitopes.
6. It's unclear to me what the hypothesis is to dissect "the factors associated with the magnitude of increased autoantibody reactivities in COVID19 patients compared to vaccinated patients". Isn't the vaccine (which codes for a small peptide) vs. the actual infection (a new virus with many different

antigens, and indeed causes cell and tissue damage and releasing of self-antigens) the reason why they result in different autoantibodies?

7. The only meaningful data in this paper is the REAP assay done on the myocarditis patients. This is the only cohort that has clinical diseases altered by the vaccination. However, the disease has such a specific target organ indicates that there should be specific functional autoantibodies, if the disease is indeed caused by autoantibodies. There is insufficient evidence to make the conclusion that "changes in autoantibodies specific for extracellular antigens are unlikely to underlie mRNA vaccine associated myocarditis". The myocarditis might very well be caused by a specific autoantibody, with high affinity but not necessarily high concentration, that cannot be identified by the REAP assay. Is there evidence showing that the REAP assay can identify functional (functional meaning blocking or neutralizing, not only binding) autoantibodies in other myocarditis patients? They will be the positive control for this experiment. I suggest the authors to focus on this cohort (all the other 7 cohorts can be used as controls), and search for functional autoantibodies targeting the tissues/cells damaged in myocarditis patients, instead of using a broad assay, without functional validation. It will be a great discovery.

Reviewer #3 (Remarks to the Author):

This is an interesting study suggesting that SARS-CoV2 infection elicits the production of antibodies while the vaccines targeting the same antigen drives a humoral autoimmunity selectively against the SARS-CoV-2 S1 receptor binding domain (SARS-CoV-2-RBD).

I have some concerns about the different time-points used in the three cohorts and whether this could have affected the results.

One of the aspects that was in the aim of the Authors to investigate regards whether the autoantibody responses after vaccination differ in individuals previously infected with

COVID-19 relative to naive individuals. The authors observe that subjects have an increase or rather a mild increase, except for few subjects who were negative at T0 who showed a marked increase (Yale HCW). How was defined SARS-CoV-2 previous infection? Why these baseline differences? How much prior to the study did these subjects suffer from COVID-19?

The responses in the csDMARDs group seem to predict the lack of production of SARS-CoV-2-RBD in two patients. Details on type of csDMARD would be interesting (maybe mycophenolate?).

At the same extent anti-TNF drugs are often used in combination with DMARDs, this should be clarified and better specified.

There is no mention on steroid treatment that may have affected the obtained results. It should be detailed if any autoimmune patients were treated with a dosage of GCs > 10 mg/day.

Moreover, if GCs were used in COVID-19 patients, could the Authors state that this treatment had no effect on the generation of autoantibodies related to the infection?

The authors should briefly discuss the potential pathogenic mechanism of vaccine-related myocarditis alternative to generation of autoantibodies specific for extracellular antigens.

REVIEWER COMMENTS

Reviewer #1 (Remarks to the Author):

The article by Jaycox and al., uses REAP (Rapid Extracellular Antigen Profiling) to study antibody responses after vaccination with SARS-CoV2 mRNA vaccines.

They study relatively large cohorts of 145 healthy vaccinated individuals, in addition to 38 patients with autoimmune diseases, and 8 patients with mRNA vaccine-associated myocarditis.

The paper is interesting and relatively novel in the question posed, and the study of post-vaccine auto-Abs, as well as auto-Abs in the context of vaccine induced myocarditis.

We thank the reviewer for appreciating the novelty of the paper and are glad that they found it interesting.

Comments:

- 1. It is at times a bit difficult to follow the different cohorts in the manuscript (Fig 1A is helpful though) and might benefit from simplifying the first result paragraphs where the cohorts are described sequentially.**

We thank the reviewer for this comment and have made changes to increase the flow and readability of this paragraph.

- 2. Are there any differences between Moderna and Pfizer mRNA vaccines?**

The Pfizer-BioNTech (BNT162b2) Moderna (mRNA-1273) are largely similar in mechanism and efficacy. Both contain a codon optimized mRNA transcript encoding the full-length SARS-CoV2 spike protein (pre-fusion state) enclosed in a lipid nanoparticle delivery system. Initial efficacy of both vaccines was similar, ranging between 94 to 95%^{1,2}. Minor differences between the Moderna and Pfizer vaccines include differences in dosage (100 µg vs 30 µg, respectively), interval between doses (28 days vs 21 days, respectively)³, and slightly different lipid nanoparticle formulations⁴. Overall, due to the similarity between the two vaccines, we chose to group them in our analysis. Furthermore, in our analysis, we saw no differences in the effects of the two vaccines on autoantibody dynamics. For example, in Figure 6D, we show that vaccine type is not a significant predictor of the magnitude of autoantibody increase in vaccinated patients in a multiple linear regression model.

1. Baden, Lindsey R., et al. "Efficacy and safety of the mRNA-1273 SARS-CoV-2 vaccine." *New England journal of medicine* (2020).
2. Polack, Fernando P., et al. "Safety and efficacy of the BNT162b2 mRNA Covid-19 vaccine." *New England journal of medicine* (2020).

3. Ioannou, George N., et al. "Comparison of Moderna versus Pfizer-BioNTech COVID-19 vaccine outcomes: a target trial emulation study in the US Veterans Affairs healthcare system." *EClinicalMedicine* 45 (2022): 101326.
4. Risma, Kimberly A., et al. "Potential mechanisms of anaphylaxis to COVID-19 mRNA vaccines." *Journal of Allergy and Clinical Immunology* 147.6 (2021): 2075-2082.

3. Details of the auto-immune's disease could be detailed in the manuscript (at least to mention the number of patients per disease).

We thank the reviewer for this comment and have made changes in the text to detail the number of patients in each disease group

4. Is Crohns considered an auto-immune disease or an inflammatory disease?

We thank the reviewer for this point. Crohn's disease is an inflammatory disorder characterized by inappropriate interactions between the host immune system and the gastrointestinal microbiome. While Crohn's disease is often classified as an autoimmune condition colloquially, the disorder is more appropriately classified as an inflammatory disease, given that the driving inflammatory agent is not technically self-antigen, but rather foreign/microbial antigen¹. Additionally, Crohn's disease involves some features of immunodeficiency, including impaired innate immune response to bacteria and impaired gut barrier function².

We believe the inclusion of this disease in our "autoimmune disease" cohort is valid and valuable; Crohn's disease recapitulates several common characteristics of autoimmune conditions, including co-occurrence with other autoimmune conditions such as ankylosing spondylitis, increased incidence in westernized countries, stereotypical autoantibodies, and responsiveness to immunosuppressants (glucocorticoids and anti-TNF therapies, for example)¹. Additionally, extraintestinal inflammation at sterile sites such as the joints also occurs in many patients, suggesting that self-antigen may drive inflammatory processes as well.

1. Roda, Giulia, et al. "Crohn's disease." *Nature Reviews Disease Primers* 6.1 (2020): 1-19.
2. Casanova, Jean-Laurent, and Laurent Abel. "Revisiting Crohn's disease as a primary immunodeficiency of macrophages." *Journal of Experimental Medicine* 206.9 (2009): 1839-1843.

5. In addition, the detection of auto-Abs to TNFa is interesting but not relevant for the present study. Removing these patients could be considered.

We included therapeutic antibodies in the example line plots because this is an important internal positive control that helps to demonstrate the sensitivity of REAP to detect antibodies against the exoproteome (whether they are drugs or autoantibodies). We agree with the reviewer, however, that these antibodies are not relevant to the conclusions of the study; for this reason, we now

depict these antibodies in the line plots with a different color and have excluded likely drug antibodies (anti-TNF, anti-IL6R) from all subsequent quantitative analyses in the paper. We have updated the manuscript figure legends to make this clearer.

6. In figure 1B, C it would be helpful to split the figure in two, to really identify the vaccinated and unvaccinated individuals

To clarify, in figure 1B and 1C, all individuals are actually vaccinated. The goal of these figures was to show the CoV2-RBD REAP score change during the course of vaccination. The pre- and post- vaccination time points are denoted by “Pre” and “Post” on the x axis. Each individual has one point in the “Pre” column and one point in the “Post” column connected by a line to indicate continuity within one individual.

7. In the second paragraph, the authors should differentiate the auto-Abs that are secondary to the infection by SARS-CoV-2 to those that are pre-existing the infection and can cause severe disease (i.e. Abs to type I IFNs).

We cannot definitively say which autoantibodies in our COVID-19 study were preexisting versus newly derived because we do not have pre-infection blood samples for any of the infected individuals. These individuals were recruited and became known to us upon their admission to Yale New Haven Hospital in March to May 2020 for COVID-19. The reviewer is correct that some of the autoantibodies we detected may have been pre-existing and unrelated to SARS-CoV2 infection. We have modified the text to note this. However, we note that the analysis of autoantibodies in SARS-CoV2 infected patients in this paper focuses entirely on the *temporal dynamics* of the autoantibodies, for example measuring the number of new autoantibodies or the magnitude of increased autoantibodies, starting from the first observed time point post SARS-CoV2 infection. In this sense, the manuscript focuses on changes in the autoantibody compartment that occur soon after the onset of COVID-19, and thus are most likely to be related to infection. In our previous work, we classified these antibodies as “newly acquired” post infection and antibodies present at the first timepoint and before the development of anti-spike as “likely pre-existing.” We may be underestimating the true number of new autoantibodies in COVID-19 patients if they developed after onset of infection and prior to the first plasma timepoint that was collected.

8. It would be interesting to stratify per auto-immune disease, to see if there are common auto-Abs that persist. Also, does REAP detect the auto-Abs known to be associated to these AI diseases (especially T1D, RA...)

We agree this is interesting and direct the reviewer to our previous paper where we explore the topic (Wang et al., 2022, Cell Reports Methods. In that paper, we analyzed over 100 patients with Systemic Lupus Erythematosus using REAP to look for shared and private autoantibody reactivities in that population. It turns out that most autoantibodies we detected in SLE are relatively private, occurring in <5% of patients. So given the smaller cohort sizes of the different autoimmune diseases tested here, it would not be possible to detect many shared extracellular

autoantibodies. As for the second part of the question, most conventional disease-associated diagnostic autoantibodies are intracellular and not included in the exoproteome library (e.g., anti-citrullinated peptide antibodies in RA and anti-GAD-65 and anti-IA2 in T1D). As for extracellular antigens, the library includes sequences for IgG Fc (including IgG1-4), which is targeted by Rheumatoid Factor (RF) in RA. However, these antigens cannot be detected in current iterations of REAP that rely on magnetic separation using a secondary antibody directed against IgG Fc. Reactivity towards these antigens thus cannot be distinguished from secondary only binding.

9. In MS patients treated with Rituximab, do the authors have the information on the date of last injection of number of B cells? That would be of particular interest for the patient with a normal response.

We thank the reviewer for their question. We do have this data and have prepared the plot below to show the relationship between days from final anti-CD20 dose with CoV2-RBD REAP score at the final post-vaccination time point for each patient. Interestingly, there is a non-significant trend associating better responses to the vaccine with fewer days from last dose. While non-significant, this is somewhat counter-intuitive, as one would expect B cells levels to recover as more days elapsed from the final dose. Overall, it is difficult to draw conclusions from this small group of patients.

10. How can the authors explain a good response by ELISA or REAP titers, but a decreased neutralization capacity, in figure 2E?

We assume the reviewer is referring to the S1 ELISA and SARS-CoV2 neutralization data presented in figure 1E /1F, as opposed to 2E which is focused on autoantibodies. We agree with the reviewer that this is an interesting, if not somewhat paradoxical finding. We believe this reflects a constrained humoral response due to the patients' immunosuppressive therapies, resulting in the targeting of non-neutralizing or less potently neutralizing regions of the Spike protein. We now address this in the discussion, writing:

“Notably, we found that although most autoimmune patients generate RBD-specific antibodies after vaccination, their antibodies were frequently non-neutralizing, unlike healthy individuals where the correlation between antibody titers and neutralizing ability was strong. This suggests that the anti-SARS-CoV-2 antibody response generated in these individuals may be constrained and less effective, for example due to targeting of non-critical epitopes or lack of protective anti-N-terminal-domain (NTD) antibodies¹⁷. This was particularly evident for individuals on B cell depletion, who were the least likely to generate a neutralizing antibody response, in accordance with another recent report¹⁸. Interestingly, some of these patients nevertheless generated non-neutralizing antibodies against the full S protein, which likely reflects an inefficient humoral response constrained by B cell depletion.”

11. How would the authors explain fig 1G, showing a cross-reactivity to non-RBD epitopes in individuals that have only been vaccinated by mRNA vaccines against S protein? Sequence homology? Has this been described elsewhere?

Reactivity to non-RBD epitopes are expected after immunization with the SARS-CoV2 mRNA vaccines, which both contain mRNA encoding the full length S protein. The S protein is comprised of multiple domains, including the receptor binding domain (RBD), but also the N-terminal domain (NTD), and within S2, the fusion peptide (FP), transmembrane domain (TM), and cytoplasmic tail (CT), among others¹. While antibodies targeting the RBD generally confer the bulk of immune protection and neutralization, antibodies directed at non-RBD regions, for example the NTD and S2, are also reported after vaccination². NTD directed antibodies have been reported to be both neutralizing and non-neutralizing^{2,3}.

1. Wrapp, Daniel, et al. "Cryo-EM structure of the 2019-nCoV spike in the prefusion conformation." *Science* 367.6483 (2020): 1260-1263.
2. Amanat, Fatima, et al. "SARS-CoV-2 mRNA vaccination induces functionally diverse antibodies to NTD, RBD, and S2." *Cell* 184.15 (2021): 3936-3948.
3. Chi, Xiangyang, et al. "A neutralizing human antibody binds to the N-terminal domain of the Spike protein of SARS-CoV-2." *Science* 369.6504 (2020): 650-655.

12. For the auto-Abs arising after COVID-19 infection, have the authors assessed their persistence over time?

We thank the reviewer for this interesting question. We are currently undertaking a follow-up study to serially monitor autoantibody persistence in patients recruited to our acute COVID19 cohort. Preliminarily from a limited number of patients, we have found that some but not all autoantibodies persist several months after infection. However this study is still ongoing and we are working to obtain additional follow-up samples so we can make stronger conclusions.

13. Do the authors find auto-Abs to type I IFNs in the patients infected with severe COVID-19. If yes, do they also detect them before infection?

Yes, in accordance with previous reports¹, our acute COVID19 cohort was characterized by an elevated prevalence of anti-T1 IFN autoantibodies. 5/23 patients in the severe cohort and 2/36 patients in the moderate cohort had T1 IFN autoantibodies (see heatmap below). 3 patients in the severe group had pan-binding IFN α reactivities. In our previous paper, we established that these antibodies neutralized IFN signaling *ex vivo*. Unfortunately, we did not have pre-infection time point samples for patients in this cohort, as they were recruited during their hospitalization for acute COVID19. The T1 IFN reactivities observed were largely stable over time (see line plot below). Furthermore, for some patients anti-IFN α autoantibodies were detected as early as day 8 post symptom onset and prior to the development of anti-SARS-CoV-2 RBD antibodies. Conservatively, the IgG producing plasmablast response takes at least 5 days to begin, while production of IgG from bone marrow plasma cells takes even longer, not beginning until 11 - 14 days². Given the stability of these antibodies over time and their presence at early time points, we believe that these autoantibodies were likely preexisting. This is consistent with the conclusions of Dr. Casanova's group and others.

Heatmap of T1 IFN pathway reactivities. Left block: moderate COVID19; Right block: severe COVID19

Line plot of T1 IFN reactivities over time in the COVID19 cohort. Each color represents reactivities from one patient. Each line represents one reactivity.

1. Bastard, Paul, et al. "Autoantibodies against type I IFNs in patients with life-threatening COVID-19." *Science* 370.6515 (2020): eabd4585.
2. Weisel, Florian J., et al. "A temporal switch in the germinal center determines differential output of memory B and plasma cells." *Immunity* 44.1 (2016): 116-130.

14. For patients with myocarditis, could some pathogenic auto-Abs be undetected by REAP, either because they are not included in the panel or because of a conformational issue? It might be important to discuss this.

In total, the REAP library includes 6,183 different human extracellular or secreted proteins/epitopes which comprise the human extracellular proteome (exoproteome). Autoantibodies targeting intracellular proteins, such as Myosin heavy chain¹, have been reported in inflammatory or infection related myocarditis. These autoantibodies would not have been detected in our study because they are intracellular proteins and not included in our library. We have modified the text to acknowledge this.

Due to the breadth of our library, it is not possible for us to validate all 6,183 proteins using an orthogonal method. Thus, it is possible that certain proteins are not displayed well in the yeast display library, and this could generate a false negative result. Despite the limitations that come with the high-throughput nature of the REAP method, we have strived to validate the REAP library as extensively as possible. The REAP method has been previously published in two separate reports^{2,3}, which both demonstrate secondary validation or recapitulation of previously described reactivities for hundreds of antigens. We also performed a receiver - operating characteristic analysis to show that REAP predicted autoantibody reactivity verified by a secondary method (ELISA or LIPS) with an area under the curve (AUC) of 0.785 (Figures S5S and S5T). However, because REAP exhibits greater sensitivity for some antigens than the ELISA/LIPS "standard" assays (for example, type I IFN autoantibodies in APS-1), this number may represent a conservative estimate of the true performance of REAP in predicting true autoantibody reactivity.

In summary, while REAP does not have perfect sensitivity and specificity for all 6,183 antigens, even at ~80% accuracy, it represents a significant advance in autoantibody screening capabilities. The application of REAP to the myocarditis cohort thus represents the broadest survey of autoantibodies in this condition to date, and our findings of a lack of extracellular autoantibody reactivity in myocarditis are therefore important and informative.

1. Bracamonte-Baran, William, and Daniela Čiháková. "Cardiac autoimmunity: myocarditis." *The Immunology of Cardiovascular Homeostasis and Pathology* (2017): 187-221.
2. Wang, Eric Y., et al. "High-throughput identification of autoantibodies that target the human exoproteome." *Cell reports methods* 2.2 (2022): 100172.
3. Wang, Eric Y., et al. "Diverse functional autoantibodies in patients with COVID-19." *Nature* 595.7866 (2021): 283-288.

15. All the patients with myocarditis had been vaccinated with Pfizer. Do the authors have an explanation, based on their cohort recruitment? It should be mentioned in the text.

We thank the reviewer for this comment. The bias toward the Pfizer vaccine in the myocarditis cohort is due to the timing of the EUA for the vaccines in those under 18 relative to the timing of our study. Specifically, the Pfizer vaccine received its EUA earlier than Moderna:

- Pfizer-BioNTech vaccine, 12 years of age and older, approval date: May 10, 2021
- Moderna vaccine, for 6 months and older, approval date: June 17, 2022
- Majority of our cases were recruited in May 2021 and June 2021

We have updated the text to note this.

16. The title is a bit ambiguous, as the reader cannot understand at a glance if there are more auto-Abs in vaccinated individuals or the opposite.

We appreciate the reviewer's feedback and will confer with the editor about making changes to the title if the paper is accepted for publication.

Reviewer #2 (Remarks to the Author):

In this manuscript, Jaybox [sic] et al. used REAP score, an assay that detects several thousands of extracellular antigens, to test the humoral responses in individuals vaccinated with mRNA vaccine, with or without autoimmune diseases, or myocarditis triggered by vaccination. They hypothesis tested in this paper is whether COVID mRNA vaccine triggers the production of autoantibodies in the recipients.

Correct.

Overall, I found the hypothesis is not clearly defined, the design of the study lacks rationale, the method and cohorts chosen are not appropriate to test the hypothesis.

We are confused by the reviewer's comment here- their statement above summarizes the study and defines its core hypothesis quite clearly. We also note that the other reviewers remarked that the paper is interesting and novel, indicating it has a sound rationale. We address the appropriateness of the REAP method and cohorts below.

1. The so called "autoimmune patients" is a massively heterogeneous cohort, including MS, SLE, T1D, etc. Lumping them together as one cohort ignores the fact that they have different autoantibodies, and different disease mechanisms.

We agree the autoimmune disease patients included are diverse and heterogeneous. This was intentional. At the time of the data collection and analysis, which occurred soon after approval of the 2-dose mRNA vaccine for all individuals, there was significant concern regarding how individuals with autoimmune disease and/or on immunosuppression, in general, would respond to the vaccine relative to healthy individuals. This focused on two main issues: 1) the theoretical risk for autoimmune disease flare; and 2) the potentially attenuated vaccine response as a result of immunosuppression¹. Instead of focusing on a single autoimmune condition, we sought a diverse cohort more reflective of population-wide autoimmune prevalence to increase the relevance and generalizability of the findings.

Furthermore, while the specifics of these autoimmune diseases are unique, all autoimmune conditions included here in general reflect common defects in tolerance to self (or self-microbiome) leading to inappropriate inflammation, and many share common susceptibility alleles and environmental factors, and respond to similar modes of immunosuppression. It is thus reasonable to hypothesize that individuals with overt autoimmune diseases could be more prone to breaches of self-tolerance during inflammation induced by a vaccine than individuals without autoimmune disease. In any case, the main goal of our study was to compare autoantibody dynamics in COVID-19 infection, SARS-CoV2 vaccination of healthy individuals, **and** SARS-CoV2 vaccination in individuals who would be *predisposed to loss of self-tolerance*. We believe our autoimmune disease cohort is an adequate population to serve as this final group, even with a mix of diseases included. The fact that we continued to observe stark differences in

autoantibody dynamics even in the autoimmune vaccinated patients relative to the COVID19 patients further bolsters the strength of our conclusions compared to comparison of only healthy vaccinated individuals with COVID19.

1. Curtis, Jeffrey R., et al. "American college of rheumatology guidance for COVID-19 vaccination in patients with rheumatic and musculoskeletal diseases: version 3." *Arthritis & Rheumatology* 73.10 (2021): e60-e75.

2. Among the autoimmune patients tested in this paper, did any of them had autoimmunity flare or change of clinical diseases post vaccination? If not, it means that the autoimmune diseases in these patients are not associated with vaccination, therefore they are not the appropriate patients to study. Just because they have "autoimmune diseases", which again is not one well-defined disease, doesn't make them the correct patients to study for autoantibodies.

There were no reports of worsening of disease in any of the patients in the cohort after mRNA vaccination.

We disagree with the reviewer's assertion that if a patient did not have clinical worsening of disease after vaccination then this must mean there is no interesting relationship between autoantibodies and vaccination. For some autoimmune diseases (SLE and T1D, for example), autoantibodies precede overt symptomatic disease by several years^{1,2}. Therefore, autoantibody changes could occur in the absence of immediate symptom changes or disease flares. These autoantibody changes, particularly if shared, could indicate an underlying phenomenon tied to vaccination that would be important to note. The fact that we did not find any such changes is important.

Second, we disagree with the reviewer's final assertion that "just because they have autoimmune diseases....doesn't make them the correct patients to study for autoantibodies." The goal of this study was to understand how the autoantibody dynamics of SARS-CoV2 vaccination contrast with that of SARS-CoV2 infection. The stark difference between these two scenarios, with vaccination leading to largely no change in autoantibodies, was observed initially in healthy individuals only. We decided to then further stress-test these conclusions in individuals who would be more prone to loss of self-tolerance: autoimmune disease patients. That we observed no differences in autoantibody dynamics between autoimmune and healthy individuals post-vaccination further bolsters our findings in the paper and expands the relevance of our study beyond healthy individuals.

1. Arbuckle, Melissa R., et al. "Development of autoantibodies before the clinical onset of systemic lupus erythematosus." *New England Journal of Medicine* 349.16 (2003): 1526-1533.
2. Bosi, Emanuele, et al. "Impact of age and antibody type on progression from single to multiple autoantibodies in type 1 diabetes relatives." *The Journal of Clinical Endocrinology & Metabolism* 102.8 (2017): 2881-2886.

3. What does day 0 mean in the longitudinal cohort? The onset of T1D?

Day 0 is the start date of the vaccination series, or the day of the first dose of vaccine.

4. The autoimmune diseases the authors studied in this paper, are not characterized by specific functional autoantibodies like anti-ACHR in myasthenia gravis. The author did not show that the REAP assay can identify the characteristic autoantibodies such as anti-beta cell in T1D. What is the rationale for using the REAP assay, which contains several thousand antigens, to test the antibody response of mRNA vaccine which contains RNA that codes for a small peptide? I understand the rationale for using REAP assay in COVID patients, who encountered a new pathogen with a variety of new antigens, and actually leads to tissue damage that releases many self-antigens and PAMPs. But for mRNA vaccine, it is not surprising that a small peptide did not change the diversity of several thousands of autoantibodies in any of the cohorts the authors tested. Is there any evidence in the literature that a small peptide can change the diversity of autoantibody/antibody pool in vivo?

[Prior to answering the questions posed by the reviewer, it is important to point out that their statement above contains a crucial error in fact related to the nature of the COVID-19 mRNA vaccines administered. They refer to the vaccine antigen as “a small peptide” but in reality, the mRNAs encode a very large multi-domain S protein (~1400 kDa). As for how and why vaccination could change the diversity of the autoantibody landscape in patients, we detailed in the manuscript concerns about the potential for molecular mimicry against the Spike protein as well as vaccine-induced systemic inflammation. Prior to our study, both possibilities were widely hypothesized to drive potential autoimmune/autoinflammatory sequelae with COVID-19 vaccination (discussed below). Hence the basis for this study.]

With respect to why REAP was used in this study, the REAP assay is an unbiased screening tool capable of detecting autoantibodies against over 6,000 extracellular proteins and epitopes. REAP allows us to comprehensively understand the breadth of autoantibody dynamics during inflammation in a completely unbiased fashion, as opposed to the traditional approaches which measure reactivity against one antigen at a time. This would allow us to detect unanticipated or even novel autoreactivity that could occur as a result of molecular mimicry to the SARS-CoV-2 S protein antigen or as a result of systemic inflammation from the vaccine. Our study likely represents the broadest survey of autoantibody responses during and after vaccination to date.

When this study was conducted in early 2021, the mRNA vaccines represented a completely novel platform with a relatively unknown safety record, particularly for individuals with preexisting inflammatory disease. Furthermore, concerns arose surrounding molecular mimicry of the human placental antigen Syncytin-1 with the Spike protein contained in the vaccine. While this has now been disproven¹, numerous studies and commentary continue to be published claiming that either spike protein molecular mimicry or the inflammatory effect of the vaccine presents risk for

autoimmunity associated with vaccination^{2,3,4}. Furthermore, we believe that the emergence of mRNA vaccine-associated myocarditis, reports of pre-existing autoimmune disease flares after vaccination^{5,6,7}, and case reports describing new onset autoimmune syndromes^{8,9,10} after vaccination all provide additional justification to examine autoantibody responses after vaccination using a high throughput tool such as REAP.

Regarding the reviewer's final question, we again note that the vaccine does not encode for a small peptide but for a large 1.4 megadalton protein. Nevertheless, it is widely accepted that certain vaccines can cause clinical syndromes that are likely to be underlined at least in part by autoantibodies. For example, Guillain Barre syndrome has been associated with vaccination against specific influenza strains and Shingles/Recombinant Zoster Vaccine (RZV), and is typically marked by anti-ganglioside autoantibodies^{11,12}. While none of these vaccines are based on the mRNA platform, they are all narrow in the diversity of antigens they contain and present a lower inflammatory profile compared to active infection. For example, none of these vaccines contain live virus, and in the case of RZV, only a single protein (glycoprotein E) plus adjuvant is delivered in the vaccine¹³. While the mechanism of this disease process is poorly understood, it is possible that in susceptible individuals, either molecular mimicry or inflammation as a result of the vaccination may activate autoreactive lymphocytes to produce autoantibodies or attack self-antigens.

1. Prasad, Mukul, et al. "No crossreactivity of anti-SARS-CoV-2 spike protein antibodies with Syncytin-1." *Cellular & Molecular Immunology* 18.11 (2021): 2566-2568.
2. Nunez-Castilla, Janelle, et al. "Potential autoimmunity resulting from molecular mimicry between SARS-CoV-2 Spike and human proteins." *Viruses* 14.7 (2022): 1415.
3. Vojdani, Aristo, and Datis Kharrazian. "Potential antigenic cross-reactivity between SARS-CoV-2 and human tissue with a possible link to an increase in autoimmune diseases." *Clinical Immunology (Orlando, Fla.)* 217 (2020): 108480.
4. Cuspoca, Andres Felipe, Pablo Isaac Estrada, and Alberto Velez-van-Meerbeke. "MOLECULAR MIMICRY OF SARS-COV-2 SPIKE PROTEIN IN THE NERVOUS SYSTEM: A BIOINFORMATICS APPROACH." *Computational and Structural Biotechnology Journal* (2022).
5. Terracina, Katherine A., and Filemon K. Tan. "Flare of rheumatoid arthritis after COVID-19 vaccination." *The Lancet Rheumatology* 3.7 (2021): e469-e470.
6. Niebel, Dennis, et al. "Exacerbation of subacute cutaneous lupus erythematosus following vaccination with BNT162b2 mRNA vaccine." *Dermatologic Therapy* (2021).
7. Izmirly, Peter M., et al. "Evaluation of immune response and disease status in systemic lupus erythematosus patients following SARS-CoV-2 vaccination." *Arthritis & Rheumatology* 74.2 (2022): 284-294.
8. Khan, Farooq, and Mary Jane Brassill. "Subacute thyroiditis post-Pfizer-BioNTech mRNA vaccination for COVID-19." *Endocrinology, Diabetes & Metabolism Case Reports* 2021.1 (2021).
9. Kaminetsky, Joshua, and Donald Rudikoff. "New onset vitiligo following mRNA-1273 (Moderna) COVID 19 vaccination." *Clinical Case Reports* 9.9 (2021): e04865.
10. Cavalli, Giulio, et al. "Cutaneous vasculitis following COVID-19 vaccination." *The Lancet Rheumatology* 3.11 (2021): e743-e744.
11. Goud, Ravi, et al. "Risk of Guillain-Barré syndrome following recombinant zoster vaccine in Medicare beneficiaries." *JAMA Internal Medicine* 181.12 (2021): 1623-1630.

12. Arias, LH Martín, et al. "Guillain-Barré syndrome and influenza vaccines: a meta-analysis." *Vaccine* 33.31 (2015): 3773-3778.
13. Stoker, Kalvin, Terri L. Levien, and Danial E. Baker. "Zoster Vaccine Recombinant, Adjuvanted." *Hospital Pharmacy* 53.3 (2018): 136-141.

5. How is it possible that some patients generated non-RBD antibodies through vaccination (Fig 1G)? I think the better explanation for this data are: (1) these patients had COVID before and vaccination failed, therefore they have antibodies against S but not RBD; or (2) which is more likely, that the RBD-ELISA does not detect all anti-RBD antibodies, which is very common in ELISA-based assay especially with small peptides, as the coating of the peptide to the ELISA plates can selectively mask some epitopes.

We again note that the reviewer's question contains an error in fact about the nature of the SARS-CoV-2 mRNA vaccines, which do not encode "small peptides" but a ~1400 kDa Spike protein. Reactivity to non-RBD epitopes is thus expected after immunization with the SARS-CoV2 mRNA vaccines. The Spike protein is comprised of multiple domains, including the receptor binding domain (RBD), but also the N-terminal domain (NTD), and within S2, the fusion peptide (FP), transmembrane domain (TM), and cytoplasmic tail (CT), among others¹. While antibodies targeting the RBD generally confer the bulk of immune protection and neutralization, antibodies directed at non-RBD regions, for example the NTD and S2, are also reported after vaccination². NTD directed antibodies have been reported to be both neutralizing and non-neutralizing^{2,3}.

The reviewer's speculation on possible explanations for our data are based off erroneous assumptions of the composition of the mRNA vaccines as "small peptides". However, in reference to hypothesis #1, we note that none of the autoimmune cohort individuals reported symptomatic or PCR/rapid test verified COVID19 prior to vaccination. While prior asymptomatic infection is theoretically possible, it is less likely, given that most of these individuals are immunocompromised.

Secondly, we disagree with the reviewer's hypothesis #2, which asserts that the ELISA is not detecting certain epitopes of the RBD (which for the record is not a "small peptide" either but a 222 amino acid 27 kDa protein). The ELISA technique used for this study has been validated and published numerous times^{4,5,6,7}. ELISA is the gold standard assay for monitoring SARS-CoV2 humoral immunity. The reviewer's assertion that there could be "masking" of certain epitopes by the ELISA plates is speculative and contrary to extensive literature using ELISA in this context. Furthermore, the strong correlation between the SARS-CoV2 RBD ELISA titer and the CoV2-RBD REAP score supports the validity of both techniques (Fig. S1D). Of particular relevance to the reviewer's concern, we would like to highlight that the detection of anti-RBD reactivity by REAP relies on yeast surface display in solution and therefore would not be constrained by the coating of the peptide to an ELISA plate. If ELISA was indeed constrained by the limitations the reviewer raised, one would expect more discordance between these two techniques, with ELISA consistently underestimating reactivity.

Taken together, we believe the most likely conclusion is the one that we put forth in the paper, which is that the abnormal humoral response to vaccination observed in immunosuppressed patients, particularly those on B cell depletion therapies, reflects an inefficient and/or constrained response to the mRNA vaccine.

1. Wrapp, Daniel, et al. "Cryo-EM structure of the 2019-nCoV spike in the prefusion conformation." *Science* 367.6483 (2020): 1260-1263.
2. Amanat, Fatima, et al. "SARS-CoV-2 mRNA vaccination induces functionally diverse antibodies to NTD, RBD, and S2." *Cell* 184.15 (2021): 3936-3948.
3. Chi, Xiangyang, et al. "A neutralizing human antibody binds to the N-terminal domain of the Spike protein of SARS-CoV-2." *Science* 369.6504 (2020): 650-655.
4. Takahashi, Takehiro, et al. "Sex differences in immune responses that underlie COVID-19 disease outcomes." *Nature* 588.7837 (2020): 315-320.
5. Lucas, Carolina, et al. "Longitudinal analyses reveal immunological misfiring in severe COVID-19." *Nature* 584.7821 (2020): 463-469.
6. Lucas, Carolina, et al. "Delayed production of neutralizing antibodies correlates with fatal COVID-19." *Nature medicine* 27.7 (2021): 1178-1186.
7. Lucas, Carolina, et al. "Impact of circulating SARS-CoV-2 variants on mRNA vaccine-induced immunity." *Nature* 600.7889 (2021): 523-529.

6. It's unclear to me what the hypothesis is to dissect "the factors associated with the magnitude of increased autoantibody reactivities in COVID19 patients compared to vaccinated patients". Isn't the vaccine (which codes for a small peptide) vs. the actual infection (a new virus with many different antigens, and indeed causes cell and tissue damage and releasing of self-antigens) the reason why they result in different autoantibodies?

Once again, we note that the reviewer is mistaken in their assertion that the vaccines encode for "a small peptide" rather than the ~1400 kDa Spike protein. In any case, prevailing hypotheses for how infections can elicit autoimmunity include tissue damage/inflammation as well as molecular mimicry, as we discussed in the manuscript. There are over 150 articles in pubmed that describe "molecular mimicry" and "SARS-CoV-2" including some that directly implicate molecular mimicry with the Spike protein.

We note that prior to our study, no one had directly compared the autoantibody responses of COVID-19 infection with SARS-CoV2 vaccination. At the time of our study, the mRNA vaccine platform was a new technology, and as mentioned above, numerous uncertainties existed regarding molecular mimicry of the Spike protein and the behavior of the vaccine in immunocompromised or autoimmune individuals. We indeed hypothesized that the autoantibody dynamics observed in vaccination would be less volatile than those observed in COVID-19, precisely due to some of the reasons the reviewer suggests: less tissue damage, less release of self-antigen, and the absence of inflammatory immunopathology. However, we were also curious

whether we would see stereotypical autoantibody responses or behavior after vaccination. The fact that we did not observe these findings is important for public health, and further underlies the conclusion that the abnormal autoantibody responses observed in COVID-19 are related to immunopathology and not molecular mimicry with the Spike protein, as has been suggested previously.

7. The only meaningful data in this paper is the REAP assay done on the myocarditis patients. This is the only cohort that has clinical diseases altered by the vaccination.

Irrespective of the reviewer's opinion on what data is meaningful or not, it is important to emphasize that identifying autoantibodies responsible for immune related adverse events was not the major objective of this paper. Again, the focus of our study was to compare autoantibody dynamics during SARS-CoV2 vaccination versus infection. The decision to study autoimmune individuals as well as individuals with *bona fide* immune related adverse events (*i.e.*, myocarditis patients) was primarily to broaden the conclusions of our paper such that they were not solely based on healthy individuals.

However, the disease has such a specific target organ indicates that there should be specific functional autoantibodies, if the disease is indeed caused by autoantibodies. There is insufficient evidence to make the conclusion that "changes in autoantibodies specific for extracellular antigens are unlikely to underlie mRNA vaccine associated myocarditis". The myocarditis might very well be caused by a specific autoantibody, with high affinity but not necessarily high concentration, that cannot be identified by the REAP assay.

The reviewer is incorrect in their assertion that our study is not sufficient to support our conclusion that "changes in autoantibodies specific for extracellular antigens are unlikely to underlie mRNA vaccine associated myocarditis". To the contrary, the assessment of autoantibodies in myocarditis using REAP likely represents the broadest autoantibody screen to date in this condition, representing over 6,000 antigen/epitopes screened. The REAP assay has been previously validated in two separate publications^{1,2}, where it has detected hundreds of autoantibodies that were either previously known to exist in specific conditions or subsequently validated by gold standard methods such as ELISA and LIPS.

Secondly, in reference to the reviewer's hypothesis that a specific autoantibody might be missed by REAP if it was low concentration in serum, we would like to note that the REAP score is quantitative and directly correlates to antibody titer (itself a function of affinity and concentration). We have previously established that the limit of detection for high-affinity antibodies (using monoclonal antibodies) is ~10 ng/mL (~10 pM).¹ We have similarly performed parallel titrations of REAP and ELISA for antigens such as IFN α and found REAP to have commensurate and in some cases greater sensitivity than ELISA to detect autoantibodies.¹

1. Wang, Eric Y., et al. "High-throughput identification of autoantibodies that target the human exoproteome." *Cell reports methods* 2.2 (2022): 100172.
2. Wang, Eric Y., et al. "Diverse functional autoantibodies in patients with COVID-19." *Nature* 595.7866 (2021): 283-288.

8. Is there evidence showing that the REAP assay can identify functional (functional meaning blocking or neutralizing, not only binding) autoantibodies in other myocarditis patients? They will be the positive control for this experiment. I suggest the authors to focus on this cohort (all the other 7 cohorts can be used as controls), and search for functional autoantibodies targeting the tissues/cells damaged in myocarditis patients, instead of using a broad assay, without functional validation. It will be a great discovery.

We have not used REAP previously to study myocarditis. This is not a prerequisite for the present study, since we have already validated REAP to be capable to detect functional autoantibodies in acute COVID-19, systemic lupus erythematosus, and APS-1/APECED. These functional autoantibodies include those that block signaling function (e.g., for cytokines such as type I IFN, IL-6, IL-18, IL-22, IL-33) or to drive ADCC (e.g., against B and T cell antigens).^{1,2} There is no reason to expect that the ability for REAP to detect autoantibodies in serum from myocarditis patients would differ from its ability to detect autoantibodies in any other disease.

Unfortunately, we did not have access to serum from non-vaccine related myocarditis patients. While this would have been a nice additional sample, we disagree that it could be considered a "positive control," and instead would be just another additional unknown sample that is largely out of the scope of our paper, as vaccine associated myocarditis may have a completely different mechanism relative to other forms of myocarditis. Overall, in addition to being difficult to acquire, we do not feel that these additional samples would have further bolstered our conclusions, which are already well supported by the data presented in the paper.

1. Wang, Eric Y., et al. "High-throughput identification of autoantibodies that target the human exoproteome." *Cell reports methods* 2.2 (2022): 100172.
2. Wang, Eric Y., et al. "Diverse functional autoantibodies in patients with COVID-19." *Nature* 595.7866 (2021): 283-288.

Reviewer #3 (Remarks to the Author):

This is an interesting study suggesting that SARS-CoV2 infection elicits the production of antibodies while the vaccines targeting the same antigen drives a humoral autoimmunity selectively against the SARS-CoV-2 S1 receptor binding domain (SARS-CoV-2-RBD).

I have some concerns about the different time-points used in the three cohorts and whether this could have affected the results.

We thank the reviewer for their comments and are glad that they found our study interesting.

- 1. One of the aspects that was in the aim of the Authors to investigate regards whether the autoantibody responses after vaccination differ in individuals previously infected with COVID-19 relative to naive individuals. The authors observe that subjects have an increase or rather a mild increase, except for few subjects who were negative at T0 who showed a marked increase (Yale HCW). How was defined SARS-CoV-2 previous infection? Why these baseline differences? How much prior to the study did these subjects suffer from COVID-19?**

To clarify, we assume here that the reviewer is referring to figure 1B and asking why some individuals in the Yale HCW cohort show an increased COV2-RBD REAP score, while others show no increase in response. These bimodal responses relate to prior infection with COVID-19. In some individuals with a history of prior COVID-19 infection and a high COV2-RBD REAP score on day 0, no further increase in COV2-RBD reactivity is observed. However, all individuals with a low or 0 COV2-RBD REAP score showed an increase in COV2-RBD reactivity by the end of the vaccine course. Some of these individuals were previously infected, while others were SARS-CoV2 naive.

Previous SARS-CoV-2 infection was defined as the presence of nucleocapsid antibodies or documented nasopharyngeal PCR positivity for SARS-CoV-2 prior to the first dose of the COVID-19 vaccines (April 2020 - December 2020). For individuals with a positive PCR result recorded, the median duration between positive PCR and day 0 of vaccination was 188.5 days (IQR 98 – 226). Unfortunately, we did not collect severity metrics for these COVID-19 infections.

- 2. The responses in the csDMARDs group seem to predict the lack of production of SARS-CoV-2-RBD in two patients. Details on type of csDMARD would be interesting (maybe mycophenolate?).**

We thank the reviewer for this comment. The two patients in the DMARD group who did not produce an antibody response to the vaccine were both diagnosed with Multiple Sclerosis and taking Fingolimod. The COV2-RBD REAP scores for these patients relative to those with MS on other treatments are detailed in Figure S1C.

3. At the same extent anti-TNF drugs are often used in combination with DMARDs, this should be clarified and better specified.

We thank the reviewer and clarify that 3 patients in our study were taking both anti-TNF α biologics and DMARDs in combination. These patients are noted in a separate group from the anti-TNF α only or DMARD only groups in all relevant figures throughout the paper.

4. There is no mention on steroid treatment that may have affected the obtained results. It should be detailed if any autoimmune patients were treated with a dosage of GCs > 10 mg/day.

We thank the reviewer for this important comment. Within the autoimmune disease cohort, 4 of the RA patients had glucocorticoids listed on their medication list around the time of the vaccination series. We present below a table with these details:

ID	Age	Race	Sex	Hispanic /Latino	Disease	Prior SARS-Cov2 infection	Pre-vaccine time point	Post-vaccine time point	Vaccine	Medications	Medication Categories	Glucocorticoids
48	49	White, Caucasian	F	no	RA	naïve	up to 1 month prior to dose 1	3 months post dose 2	Moderna	Methotrexate	Anti-TNF, DMARD	Prednisone, 5mg/day, paused during vax series
51	45	Asian	F	no	RA	naïve	up to 1 month prior to dose 1	3 months post dose 2	Moderna	Etanercept	Anti-TNF	Prednisone, 5-10 mg/day, stopped 2 weeks pre dose 1
52	45	White, Caucasian	F	no	RA	Negative	day of dose 1	2 weeks post dose 2	Moderna	Adalimumab, Leflunomide	Anti-TNF, DMARD	Prednisone, 5 mg/day ongoing
58	55	White, Caucasian	F	no	RA	naïve	up to 1 month prior to dose 1	3 months post dose 2	Moderna	Methotrexate, Prednisone	DMARD	Prednisone, 5 - 10 mg PRN ongoing

Two of the patients had received glucocorticoids (GCs) either before or after the vaccination series, while two of the patients received GCs during the vaccination series. All of the patients were taking relatively low doses (10 mg/day or under). To understand whether autoantibody dynamics differed for RA patients on recent GCs versus off GCs, we assessed the number of new autoantibodies in these groups as well as the average REAP score delta per individual. There were 0 new autoantibody reactivities for all RA patients, regardless of GC status (Exo201 antigens only). GC treated vs un-treated RA patients did not show different mean REAP score deltas (Figure 1 below).

Figure 1: Mean REAP score delta for RA patients, stratified by GC status

We have updated the text and figures to reflect this additional analysis.

5. Moreover, if GCs were used in COVID-19 patients, could the Authors state that this treatment had no effect on the generation of autoantibodies related to the infection?

Consistent with clinical practice during the early pandemic (mid 2020), the majority of COVID-19 patients in our cohort who were hospitalized in the ICU and receiving mechanical ventilation likely also received glucocorticoids. We do not have precise information regarding the number, amount, or frequency of doses. However, we believe our conclusion that COVID-19 patients display elevated frequencies of new and increased autoantibodies relative to vaccine patients is still supported by the data. If anything, glucocorticoid administration to COVID-19 patients would likely lead to an artificially decreased B cell and autoantibody response, given that one of the mechanisms of this medication is decreased B cell receptor signaling and decreased expression of immunoglobulin loci¹. Therefore, the contrast with vaccination may be underestimated by immunosuppressive therapy given to severe COVID-19 patients.

1. Franco, Luis M., et al. "Immune regulation by glucocorticoids can be linked to cell type–dependent transcriptional responses." *Journal of Experimental Medicine* 216.2 (2019): 384-406.

6. The authors should briefly discuss the potential pathogenic mechanism of vaccine-related myocarditis alternative to generation of autoantibodies specific for extracellular antigens.

We thank the reviewer for this comment. As this is a rare and therefore difficult to study phenomenon, the precise mechanism for SARS-CoV2 vaccine associated myocarditis is not completely known. However, one recent report¹ suggested the following:

1. Elevated levels of proinflammatory cytokines and chemokines such as IL-18, IL-27, CXCL9, and CXCL10, alongside innate immune cell (monocyte) activation
2. Th1 T cell differentiation bias with expansion of activated cytotoxic T lymphocytes and NK cells

Taken together, we hypothesize that in susceptible individuals, the exuberant inflammation triggered by the second dose of the mRNA vaccine may lead to stress signals, cell death or cell damage of the myocardium. Subsequently, this would trigger activation of macrophages, monocytes, neutrophils, and NK cells, all of which possess inflammatory sensors for cell death/stress. Innate immune cells, in the context of the hyperinflammatory cytokine milieu, may then activate cytotoxic T cells that recognize self-antigen in cardiac tissue, leading to the T cell infiltration observed in several studies^{1,2,3}. We have updated the text to address this discussion.

1. Won T, Gilotra NA, Wood MK, Hughes DM, Talor MV, Lovell J, Milstone AM, Steenbergen C, Čiháková D. 2022. Increased Interleukin 18-Dependent Immune Responses Are Associated With Myopericarditis After COVID-19 mRNA Vaccination. *Front Immunol* 13: 851620

2. Schneider, Julia, et al. "Postmortem investigation of fatalities following vaccination with COVID-19 vaccines." *International journal of legal medicine* 135.6 (2021): 2335-2345.

3. Verma, Amanda K., Kory J. Lavine, and Chieh-Yu Lin. "Myocarditis after Covid-19 mRNA vaccination." *New England Journal of Medicine* 385.14 (2021): 1332-1334.

REVIEWERS' COMMENTS

Reviewer #1 (Remarks to the Author):

The authors have, overall, well addressed my comments.

They replied to most of the criticism and suggestions. I would maybe suggest to add the new data that they discussed in the point by point response, as it could be interesting as supplementary data:

- MS patients treated by Rituximab (although small cohort)
- Their data on type I IFN auto-Abs, and their interesting follow-up data
- +/- preliminary data on the persistence of some of these auto-Abs

Also interesting -and important- is the addition of their negative data on anti-IL1R auto-Abs.

Reviewer #3 (Remarks to the Author):

The authors properly responded to all raised comments.

RESPONSE TO REVIEWERS' COMMENTS

Reviewer #1 (Remarks to the Author):

The authors have, overall, well addressed my comments.

They replied to most of the criticism and suggestions. I would maybe suggest to add the new data that they discussed in the point by point response, as it could be interesting as supplementary data:

- MS patients treated by Rituximab (although small cohort)
- Their data on type I IFN auto-Abs, and their interesting follow-up data
- +/- preliminary data on the persistence of some of these auto-Abs

Also interesting -and important- is the addition of their negative data on anti-IL1R auto-Abs.

We thank the Reviewer for their approval of our manuscript.

Reviewer #3 (Remarks to the Author):

The authors properly responded to all raised comments.

We thank the Reviewer for their approval of our manuscript.